# Distinct roles of XPF-ERCC1 and Rad1-Rad10-Saw1 in replication-coupled and uncoupled inter-strand crosslink repair

Ja-Hwan Seol[1], Cory Holland[1], Xiaolei Li[2], Christopher Kim[1], Fuyang Li[1,2], Melisa Medina-Rivera[3], Robin Eichmiller[3], Ignacio F. Gallardo[4], Ilya J. Finkelstein [iD] [4], Paul Hasty[1], Eun Yong Shim[2], Jennifer A. Surtees[3] & Sang Eun Lee[1,2]

Yeast Rad1–Rad10 (XPF–ERCC1 in mammals) incises UV, oxidation, and cross-linking agent-induced DNA lesions, and contributes to multiple DNA repair pathways. To determine how Rad1–Rad10 catalyzes inter-strand crosslink repair (ICLR), we examined sensitivity to ICLs from yeast deleted for *SAW1* and *SLX4*, which encode proteins that interact physically with Rad1–Rad10 and bind stalled replication forks. Saw1, Slx1, and Slx4 are critical for replication-coupled ICLR in *mus81* deficient cells. Two rad1 mutations that disrupt interactions between Rpa1 and Rad1–Rad10 selectively disable non-nucleotide excision repair (NER) function, but retain UV lesion repair. Mutations in the analogous region of XPF also compromised XPF interactions with Rpa1 and Slx4, and are proficient in NER but deficient in ICLR and direct repeat recombination. We propose that Rad1–Rad10 makes distinct contributions to ICLR depending on cell cycle phase: in G1, Rad1–Rad10 removes ICL via NER, whereas in S/G2, Rad1–Rad10 facilitates NER-independent replication-coupled ICLR.

[1] Department of Molecular Medicine, Institute of Biotechnology, University of Texas Health Science Center at San Antonio, 7703 Floyd Curl Drive, San Antonio, TX 78229-3900, USA. [2] Department of Radiation Oncology, University of Texas Health Science Center at San Antonio, 7703 Floyd Curl Drive, San Antonio, TX 78229-3900, USA. [3] Department of Biochemistry, Jacobs School of Medicine and Biomedical Sciences, University at Buffalo, State University of New York, 955 Main Street, Buffalo, NY 14203, USA. [4] Department of Molecular Biosciences, Institute for Cellular and Molecular Biology, The University of Texas at Austin, Austin, TX 78712, USA. These authors contributed equally: Ja-Hwan Seol, Cory Holland, Xiaolei Li. Correspondence and requests for materials should be addressed to S.E.L. (email: lees4@uthscsa.edu)

Bi-functional alkylating compounds covalently link the two strands of the DNA double helix together, forming inter-strand crosslink lesions (ICLs), preventing the separation of the two strands and interfering with essential DNA transactions[1]. As a result, these compounds preferentially kill proliferating cells and have been widely administered as primary chemotherapeutic treatments for numerous types of cancers[1].

In eukaryotic cells, the repair of ICLs depends on the collective actions of multiple DNA damage response and repair pathways: nucleotide excision repair (NER), translesion synthesis (TLS), and homologous recombination (HR) pathways, and operates differently depending on phase of the cell cycle (reviewed in ref. [2, 3]). In vertebrate cells, most inter-strand crosslink repair (ICLR) is coupled to replication fork blockage (replication-coupled ICLR), although ICLR still occurs in G1 at a substantial level (replication-independent or replication-uncoupled ICLR)[4–7]. In yeast, ICLR may occur during both G1 and S phase[8]. ICLR in mammalian cells also depends on the Fanconi Anemia (FA) pathway. To date, twenty-one FA genes have been identified from FA patient-derived cell lines that are hypersensitive to ICL-inducing DNA-damaging agents[9]. The FA pathway modulates DNA repair mechanisms during the resolution of DNA inter-strand cross-links (ICLs)[10]. In yeast, FA pathway is largely absent, although a subset of these components is conserved[11–13].

The yeast Rad1–Rad10 heterodimer (XPF–ERCC1 in metazoans) is a structure-specific endonuclease that plays a critical role in multiple DNA repair pathways. Rad1–Rad10 (XPF–ERCC1) was originally identified as part of the NER pathway and is essential for the removal of UV-induced lesions by nicking 5′ of the DNA lesion and triggering downstream NER events[14–16]. Rad1–Rad10 and XPF–ERCC1 also remove abasic sites and 3′ blocked ssDNA ends in the absence of AP endonucleases as an alternative, sub-pathway of long-patch base excision repair (BER)[17]. In double-strand DNA break repair, Rad1–Rad10 resolves DNA inter-mediates that contain 3′ non-homologous single-strand DNA tails in single-strand annealing (SSA) and non-allelic recombination through 3′ non-homologous tail removal[18–21]. Notably, the BER and recombination functions of Rad1–Rad10 and XPF–ERCC1 can be distinguished from their NER function, although each of these pathways relies on the complex's structure-specific endonuclease activity that recognizes the presence of 3′ ssDNA at a double-strand single-strand (ds/ss) DNA junction as a common feature. In each case specific protein partners recruit Rad1–Rad10 to the DNA intermediate and dictate the substrate specificity of the XPF–ERCC1 complex: during NER, ERCC1 interacts with XPA and directs the complex to NER substrates[22]. XPF–ERCC1 also interacts with Slx4 in higher eukaryotes to mediate ICL unhooking[23, 24]. In yeast, Rad14 (the yeast XPA ortholog) recruits Rad1–Rad10 to NER substrates[25–27], whereas Saw1 directs Rad1–Rad10 to 3′ flaps by physical interaction in HR[28].

Both XPF–ERCC1 and Rad1–Rad10 have been proposed to incise 5′ to the ICL lesion and initiate the unhooking step of ICLR analogous to their roles in NER. Consistent with this premise, rad14Δ NER defective yeast cells are extremely sensitive to ICL to a level indistinguishable from that in rad1 or rad10 deleted cells. Surprisingly, however, mammalian cells deficient in XPA (the homolog of RAD14), which recruits XPF–ERCC1 to the 5′ junction at photoproducts during NER, is only mildly sensitive to ICL damage[29]. Furthermore, an ERCC1 mutation that disrupts the interaction of XPF–ERCC1 complex with XPA does not confer sensitivity to ICL[30, 31]. Most recently, FA-causing XPF mutants were shown to be deficient in ICLR but proficient in NER, an indication of separation of function with respect to DNA repair pathway[32, 33]. In vertebrate cells, the FA pathway operates in conjunction with replication and enables ICL unhooking by XPF–ERCC1[34]. The available evidence thus indicates that the NER pathway is dispensable for ICL unhooking in replication-dependent ICLR in higher eukaryotes[29] and that XPF–ERCC1 has activity in ICLR that is distinct from its role in NER. Consistent with this, XPF–ERCC1 has been implicated in the 3′ flap removal during replication fork re-establishment that follows initial unhooking steps, a non-NER step in ICLR[35].

In budding yeast, non-NER contributions of Rad1–Rad10 to ICLR have not been defined, which led to suggestions that ICLR functions differently in lower eukaryotes. However, we hypothesized that there are non-NER steps in yeast ICLR that require Rad1–Rad10 but that they are masked by the more dominant functions of NER. We thus set out to establish a genetic system in which Rad1–Rad10's non-NER activities in ICLR might be revealed. Here, we provide genetic evidence that Rad1–Rad10 has non-NER roles in replication-coupled ICLR. We also identified single amino acid point mutations in the RAD1 gene that selectively impair 3′ NHTR and non-NER functions of the endonuclease while NER remains intact. These mutations compromise Rad1–Rad10's interaction with the single-strand DNA-binding RPA protein complex, which in turn impacts Rad1–Rad10 catalytic activity in vitro. Finally, we provide evidence that analogous XPF mutations are deficient in interacting with Rpa1 and Slx4 and result in reduced recombination between dispersed repeat sequences but proficient in UV lesion repair. The results suggest that non-NER function in ICLR is conserved from yeast to human and yeast will provide a tractable system with which to dissect this critical repair pathway.

## Results

**Rad1–Rad10–Saw1 functions redundantly with Mus81 in ICLR.** Specific binding partners are necessary to recruit Rad1–Rad10 to distinct DNA intermediates. Saw1 directs Rad1–Rad10 to 3′ NHTR intermediates, whereas Rad14 directs the endonuclease to NER intermediates[25, 28, 36]. The differences allowed us to ask if there are distinct NER and non-NER activities of Rad1–Rad10 in ICLR by establishing the relative contribution of RAD1, RAD14, and SAW1 to replication-coupled and replication-independent ICLR.

To assess ICLR at different cell cycle stages, we arrested cells in G1 with α-factor and then either immediately exposed them to the $HN_2$ cross-linking agent (to test replication-independent ICLR in G1) or released them into S phase (to test replication-coupled ICLR in S/G2) before exposure to the drug. The sensitivity profile was then compared to highlight ICLR at G1 and S/G2 phases of the cell cycle (Fig. 1, Supplementary Fig. 1c-h). Cell cycle arrest was confirmed by flow cytometry with propidium iodide staining (Supplementary Fig. 1a, b).

Similar to previous reports[37, 38], cells treated in G1 exhibited increased sensitivity to $HN_2$ treatment and their survival was independent of Rad52 (Supplementary Fig. 1c). In contrast, survival of cells treated in S/G2 was heavily dependent on recombination; deletion of RAD52 greatly sensitized cells to $HN_2$ treatment (Fig. 1a). Therefore, the sensitivity to ICLs following release from G1 is a reliable readout for replication-coupled repair in yeast.

We then examined the viability of RAD1, RAD14, or SAW1-deleted cells upon ICL agent treatment at G1 and S/G2. We found that RAD1 and RAD14-deleted cells were severely sensitive to $HN_2$ at both G1 and S/G2 (Fig. 1b, Supplementary Fig. 1d). In stark contrast, SAW1-deleted cells were only mildly sensitive to $HN_2$, or cisplatin (CDDP) treatment in S/G2; these cells exhibited no sensitivity in G1 (Fig. 1a, Supplementary Fig. 1c, i). The severe ICL sensitivity of rad1Δ and rad14Δ cells thus likely reflect Saw1-independent functions related to NER intermediates, catalyzing incision 5′ to ICLs[39].

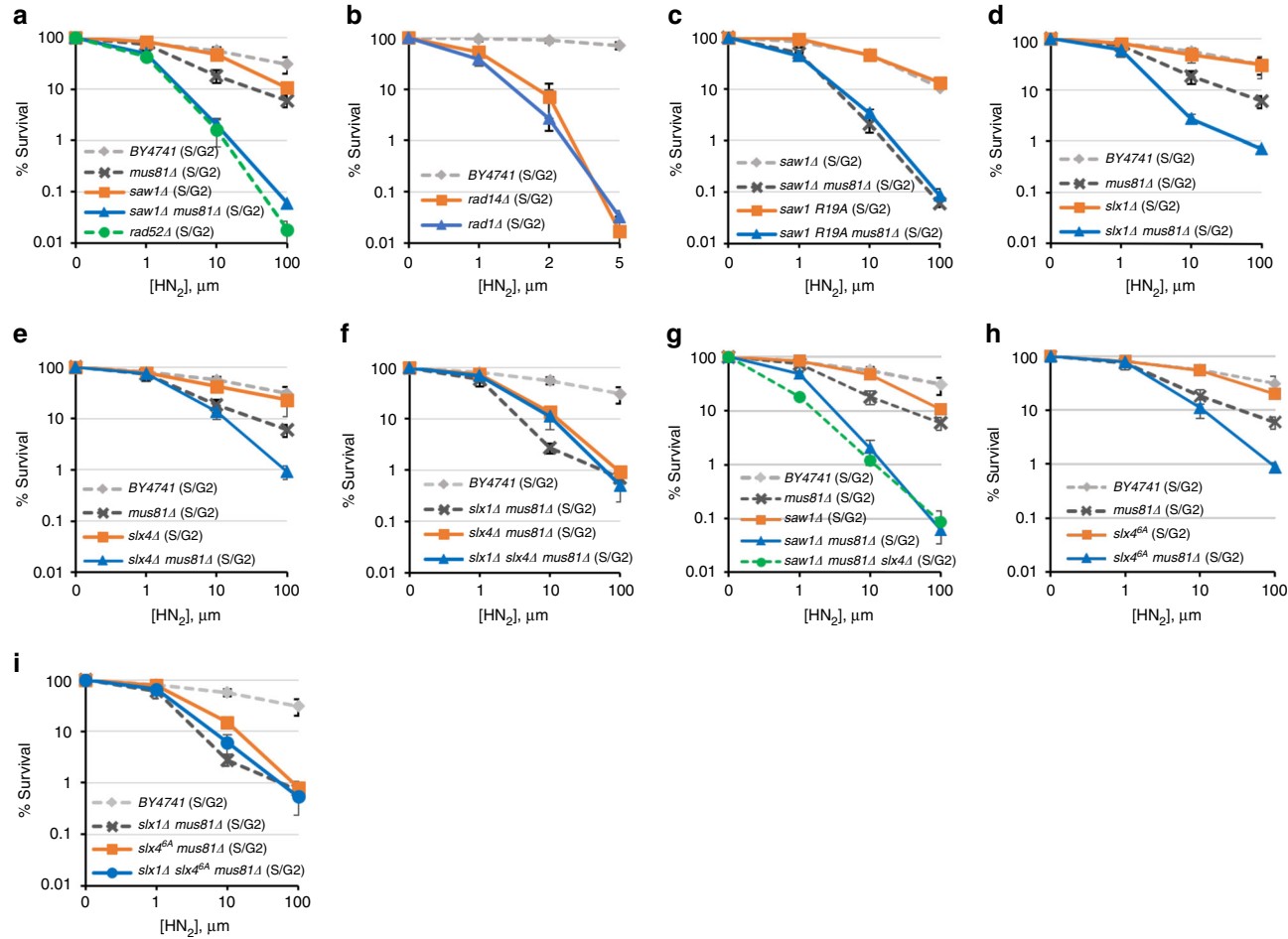

**Fig. 1** ICLR in S/G2 phase. **a-i** Yeast cells with indicated gene deletions were synchronized in G1 by α-factor treatment and then released into medium with different concentrations of HN₂ for 2 h before plating. The results of three independent experiments + s.d. are shown. Please note that the range of the x-axis in panel **b** is much smaller than all other panels, due to the extreme sensitivity of these mutants to HN₂ treatment

The lack of a detectable effect of *SAW1* deletion on ICL sensitivity prompted us to consider whether yeast cells possess a pathway that compensates for the loss of Rad1–Rad10–Saw1's activity in replication-coupled ICLR. Mus81-Mms4 is a 3′ flap endonuclease with biochemical functions most similar to those of Rad1–Rad10 for HR events in both yeast and mammals[40]. We thus deleted *MUS81* in *saw1Δ* cells and tested their ability to survive HN₂ treatment. Cells deleted for *MUS81* showed mild but reproducible sensitivity to HN₂ (Fig. 1d) and CDDP (Supplementary Fig. 1i) in S/G2, similar to the *saw1Δ* phenotype. More importantly, the *mus81Δ saw1Δ* cells showed severe sensitivity to HN₂ in S/G2, indicating that Rad1–Rad10–Saw1 and Mus81 were indeed functionally redundant for survival following HN₂ treatment (Fig. 1a). Consistent with a role specifically in replication-coupled ICLR, *mus81Δ saw1Δ* cells arrested at G1 or growing asynchronously exhibited a more moderate sensitivity to HN₂ (Supplementary Fig. 1c, j). Cells carrying *saw1-R19A*, a saw1 mutant deficient in interaction with Rad1 also showed severe sensitivity to HN₂ in *mus81Δ* cells[28] (Fig. 1c, Supplementary Fig. 1e). The HN₂ sensitivity of *saw1Δ mus81Δ* cells were comparable to that of *rad52Δ* cells (Fig. 1a). Combined these results suggest that Rad1–Rad10–Saw1 and Mus81-Mms4 contribute to most ICLR during S/G2 (replication-coupled ICLR) during which HR is critical.

**Slx1 and Slx4 function in replication-coupled ICLR.** Rad1–Rad10 (XPF–ERCC1), Mus81-Mms4 (Mus81-Eme1), and

Slx1 all interact with Slx4, a scaffold for multiple nucleases involved in ICLR and recombination[41–44]. We tested the sensitivity of *slx1Δ* and *slx4Δ* cells to HN₂ in S/G2. Cells deleted for *SLX1* or *SLX4* alone did not exhibit sensitivity to HN₂ (Fig. 1d, e, Supplementary Fig. 1f, g). However, deletion of either *SLX1* or *SLX4* in *mus81Δ* resulted in ICL sensitivity greater than the single mutants at S/G2 but not at G1 (Fig. 1d, e, Supplementary Fig. 1f, g), although not quite as sensitive as *saw1Δ mus81Δ* (compare Fig. 1a with 1d-f). The results indicate that Slx1 and Slx4 contribute to replication-coupled ICLR, and their function appears to be in a pathway distinct from that of Mus81-Mms4. The effect of *slx1Δ* and *slx4Δ* in the *mus81Δ* background appears epistatic; *slx1Δ slx4Δ mus81Δ* cells exhibited sensitivity similar to either *slx1Δ mus81Δ* or *slx4Δ mus81Δ* (Fig. 1f). The triple mutant, *saw1Δ mus81Δ slx4Δ* was essentially no more sensitive to HN₂ than *saw1Δ mus81Δ* (Fig. 1g, Supplementary Fig. 1h). We concluded that Slx1, Slx4, and Saw1 operate in the same replication-coupled ICLR pathway, but perhaps with not completely overlapping functions.

Both Saw1 and Slx4 stimulate Rad1–Rad10 nuclease activity in 3′ NHTR of branched DNA intermediates in ectopic recombination or SSA[28, 36, 45]. Importantly, the recombination-specific function of Slx4 is dependent on phosphorylation of six amino acid residues by Tel1/Mec1, the yeast homologs of ATM/ATR, respectively[45]. We expressed an Slx4 mutant with six known Mec1/Tel1 phosphorylation sites mutated to Ala (slx4⁶ᴬ) in *mus81Δ* cells and examined HN₂ sensitivity. Cells expressing slx4⁶ᴬ were sensitive to HN₂ to a level indistinguishable from that

of $slx4\Delta$ (Fig. 1h). With respect to $HN_2$ sensitivity, $slx4^{6A}$ was epistatic to $slx1\Delta$ in the $mus81\Delta$ background (Fig. 1i). We conclude that Slx4 contributes to replication-coupled ICLR by modulating the branched DNA cleavage activity of Rad1–Rad10.

**Saw1 forms ICL-induced foci.** Our genetic data indicated that Rad1–Rad10–Saw1 contributes to replication-coupled ICLR. To gain insight into how it contributes, we determined whether Saw1-GFP forms nuclear foci upon $HN_2$ treatment in S/G2[46, 47], as a marker for Rad1–Rad10–Saw1 localization because Saw1 is thought to be constitutively in complex with Rad1–Rad10; deletion of *RAD1* destabilizes Saw1 protein[28, 36, 48]. Saw1-GFP was fully functional in catalyzing SSA between *ura3* repeats flanking the HO break[28] (Supplementary Fig. 2a). The level of Saw1-GFP expression was also indistinguishable from that of untagged Saw1 (Supplementary Fig. 2b). After $HN_2$ treatment, Saw1-GFP formed distinct nuclear foci and the number of cells bearing Saw1 foci increased gradually up to a maximum of ~25% at 2 h post treatment (Fig. 2a, b). Saw1 also formed nuclear foci upon CDDP treatment (Supplementary Fig. 3), but not upon $HN_1$ treatment, which exclusively forms DNA mono-adducts (Fig. 2c)[38]. Expression of a DNA binding-deficient saw1 derivative fused to GFP (saw1$^{DB}$-GFP) did not form $HN_2$-induced nuclear foci (Fig. 2a)[28]. Saw1-GFP foci formed preferentially in S/G2 cells (Supplementary Fig. 3a, b).

HR is an integral part of ICLR in replicating cells[49]. We predicted that ICL-induced Saw1-GFP foci would mark the sites of HR. To test this idea, we co-expressed Rad52-mRFP and Saw1-GFP fusion proteins and monitored their co-localization at 2 h post $HN_2$ treatment (Fig. 2b, c). As expected, $HN_2$ treatment efficiently induced one or more Rad52-mRFP foci in ~50% of cells (Fig. 2b, c). Only cells with a bud (S/G2 cells) showed Rad52 foci. Surprisingly, $HN_2$-induced Saw1-GFP foci did not co-localize with Rad52-mRFP foci, whereas methyl methane sulfonate (MMS) treatment led to co-localization (Fig. 2b, c, GFP/RFP ratio)[50]. The results suggest that $HN_2$ induces Saw1-GFP foci at sites distinct from Rad52-bound recombination intermediates.

**Saw1 is recruited to sites of stressed replication forks.** To further investigate the relationship between Rad52 and Saw1 in replication-coupled ICLR, we used time-lapse microscopy to monitor the kinetics of Saw1-GFP and Rad52-mRFP foci in cells arrested at G1, treated with $HN_2$, then released to media lacking both α-factor and $HN_2$ (Fig. 3a). Interestingly, we found that most Saw1 foci appeared prior to Rad52-mRFP, but at distinctly different positions within the nucleus (Fig. 3a). Less than 13% of cells had Rad52-mRFP foci form prior to Saw1-GFP foci (Fig. 3b). We interpreted these results to mean that Rad1–Rad10–Saw1 acts prior to Rad52 in replication-coupled ICLR.

Previously, we showed that Saw1 binds to replication fork-like structures and facilitates cleavage by Rad1–Rad10 in reconstituted nuclease activity assays with purified proteins and fork mimicking DNA substrates[28]. ICLs trigger replication fork stalling;[51] Mrc1 associates with the stressed replication fork to form a replication

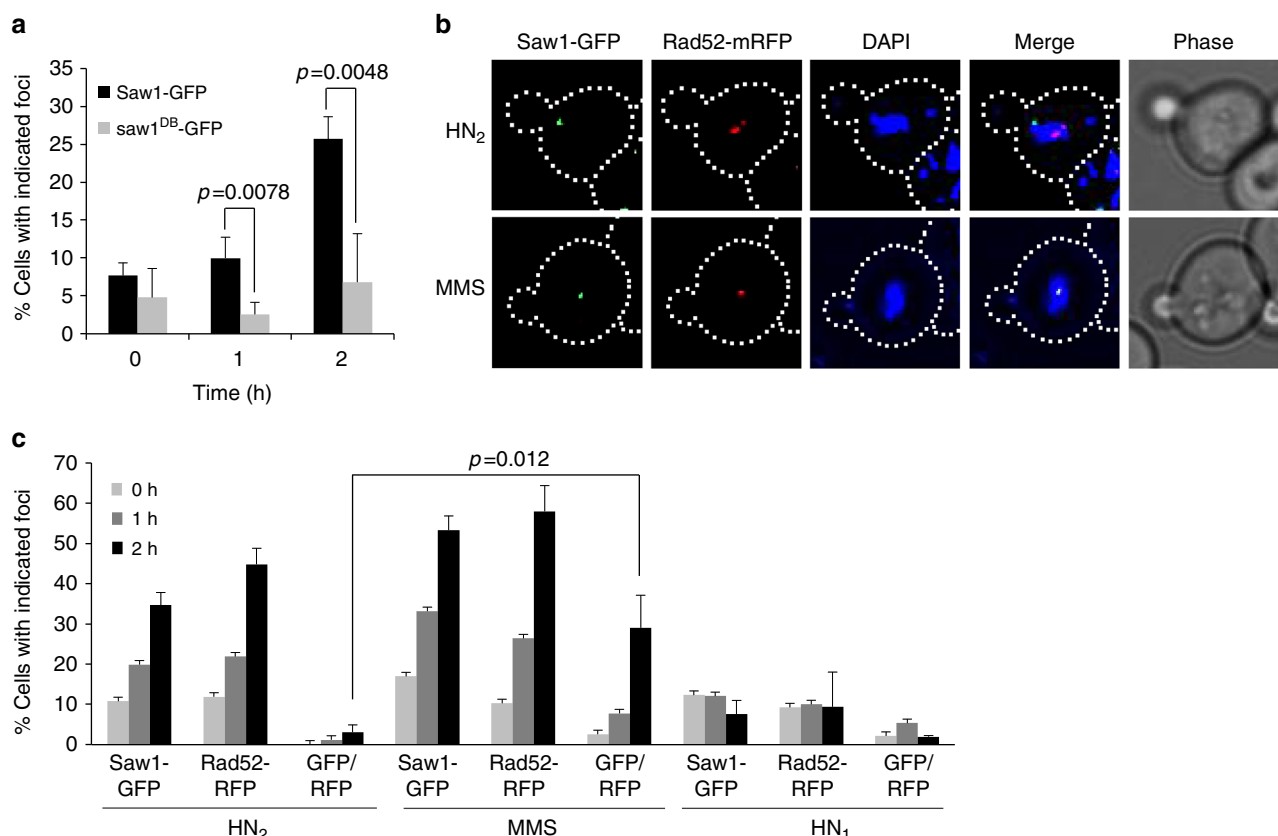

**Fig. 2** $HN_2$ treatment induces Saw1-GFP foci. **a** $HN_2$-induced Saw1-GFP and saw1$^{db-}$-GFP (DNA binding-deficient saw1 variant) foci formation. Saw1-GFP but not saw1$^{DB}$-GFP make nuclear foci upon $HN_2$ treatment (1 h, *p* value < 0.0078, 2 h, *p* value < 0.0048 using student *t*-test). **b** (Top) representative images of cells bearing $HN_2$-induced Saw1-GFP and Rad52-mRFP. (Bottom) images of MMS-induced Saw1-GFP and Rad52-mRFP. DIC images are shown for cell shape. **c** Percentage of cells bearing Saw1-GFP or Rad52-mRFP upon indicated drug treatments. Shown are the average + s.d. of three independent experiments. At least 100 cells were counted per experiment. Saw1-GFP and Rad52-GFP co-localize upon MMS but not on $HN_2$ treatment (*p* value < 0.012 using student *t*-test)

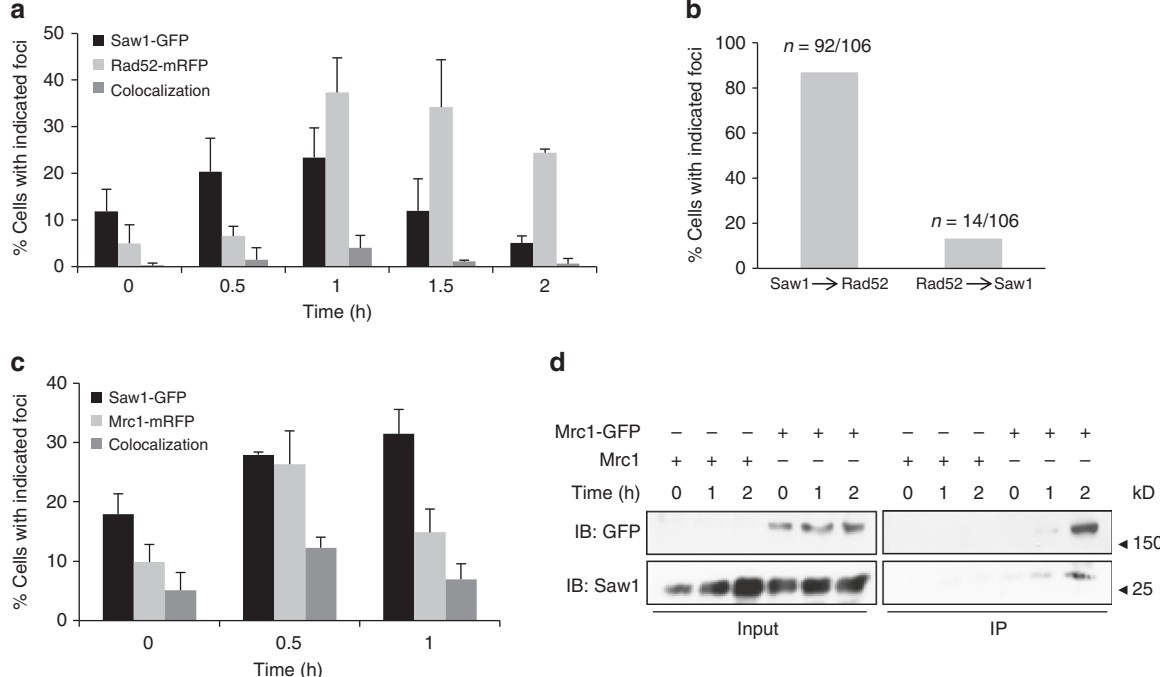

**Fig. 3** Association of Saw1 at stalled replication fork upon HN$_2$ treatment. **a** Saw1-GFP and Rad52-mRFP foci formation in cells released from G1 arrest at indicated time points post HN$_2$ treatment. Over 1000 individual cells were analyzed for nuclear foci formation. Error bars represent standard error of the mean from three independent experiments. **b** Percentage of cells forming both Saw1-GFP and Rad52-mRFP foci with a distinct temporal fashion. **c** Formation and co-localization of Saw1-GFP and Mrc1-mRFP foci at indicated time points post HN$_2$ treatment. **d** Saw1 associated with chromatin pulled down by Mrc1-GFP in cells released from G1 and at indicated time post HN$_2$ treatment

pausing complex and to transduce the signal to activate Rad53 checkpoint kinase[52, 53]. Mrc1 thus marks the site of stalled replication forks; in the absence of DNA damage, very little Mrc1 is associated with the DNA. We examined co-localization of Saw1-GFP and Mrc1-mRFP. We found that Saw1-GFP co-localized with Mrc1-mRFP upon HN$_2$ treatment, presumably at stalled replication forks (Fig. 3c). Furthermore, Saw1 is present in chromatin fractions pulled down by Mrc1-GFP after HN$_2$ treatment (Fig. 3d, Supplementary Fig. 13). Together these results suggest that Saw1 is recruited to sites of stressed replication forks following ICL damage.

**Identification of Rad1 separation of function mutations**. Our genetic and localization data suggest that Rad1–Rad10–Saw1 functions in replication-coupled ICLR are distinct from Rad1–Rad10 activity in NER. To test this hypothesis further, we screened for *rad1* separation of function mutants that are deficient in non-NER and recombination functions (dependent on Rad1–Rad10–Saw1), but retain an ability to repair UV-induced DNA lesions (dependent on Rad1–Rad10–Rad14). A yeast shuttle vector expressing *RAD1* from a constitutive *ADH1* promoter was subjected to hydroxylamine-induced, random chemical mutagenesis and was transformed into yeast cells that were deleted for both *RAD1* and *APN1* (Supplementary Fig. 4a). Yeast cells lacking *APN1* rely on *RAD1* (presumably via its Rad1–Rad10–Saw1 activity) to repair MMS-damaged bases as a 3′ flap endonuclease (Supplementary Fig. 4b)[20]. Deletion of *APN1* does not impact UV survival in wild-type or *rad1*Δ cells (Supplementary Fig. 4b). Therefore, the sensitivity of *apn1*Δ *rad1*Δ cells to MMS or UV treatment in the presence of *rad1* mutations serves as a reliable assay to monitor the integrity of Rad1–Rad10–Saw1-dependent activity (MMS) and Rad1–Rad10–Rad14-dependent activity (UV) of mutant rad1 proteins, i.e., non-NER versus NER functions. Approximately 10,000

colonies expressing rad1 mutants were screened for UV and MMS sensitivity, and the two transformants showing the most severe MMS sensitivity while retaining resistance to UV light were recovered (Supplementary Fig. 4). Sequencing revealed that these two mutants possess single amino acid substitutions at E349 or E706 to lysine in poorly defined but highly conserved regions of Rad1 and XPF (Fig. 4a).

**rad1-E349K and rad1-E706K are deficient in 3′ NHTR**. To determine whether cells carrying the newly identified *rad1* mutations perform recombination that requires 3′ NHTR, yeast strains expressing the rad1 mutants from the endogenous *RAD1* genomic locus were constructed and subjected to multiple recombination assays. First, we tested the impact of the mutations on SSA. Rad1–Rad10–Saw1 catalyzes the removal of 3′ NHTs that are key intermediates formed by annealing of flanking direct repeat sequences at DSBs in SSA[28, 36, 54, 55]. Both rad1-E349K and rad1-E706K strains exhibited severely reduced SSA between 205-bp repeats flanking the HO break (Fig. 4b) or *lacZ* repeats flanking a *Bsu*36I site on a yeast centromeric plasmid (Supplementary Fig. 5). We then examined the integrity of Saw1 and Slx4-dependent 3′ NHTR nuclease activity of rad1 mutants using a gene conversion assay that measures the removal of one or two 3′ NHTs of various sizes upon induction of an HO-induced DSB in one of the two inverted *lacZ* repeats (Supplementary Fig. 6)[36]. We found that both *rad1-E349K* and *rad1-E706K* strains are as deficient as *rad1*Δ in gene conversion between *lacZ* repeats (Supplementary Fig. 6). In contrast, *rad1-E349K* or *rad1-E706K* strains were resistant to even very high doses (>50 J/m$^2$) of UV treatment (Fig. 4c). These results confirmed that the mutations are defective in recombination and BER, but largely proficient in UV repair.

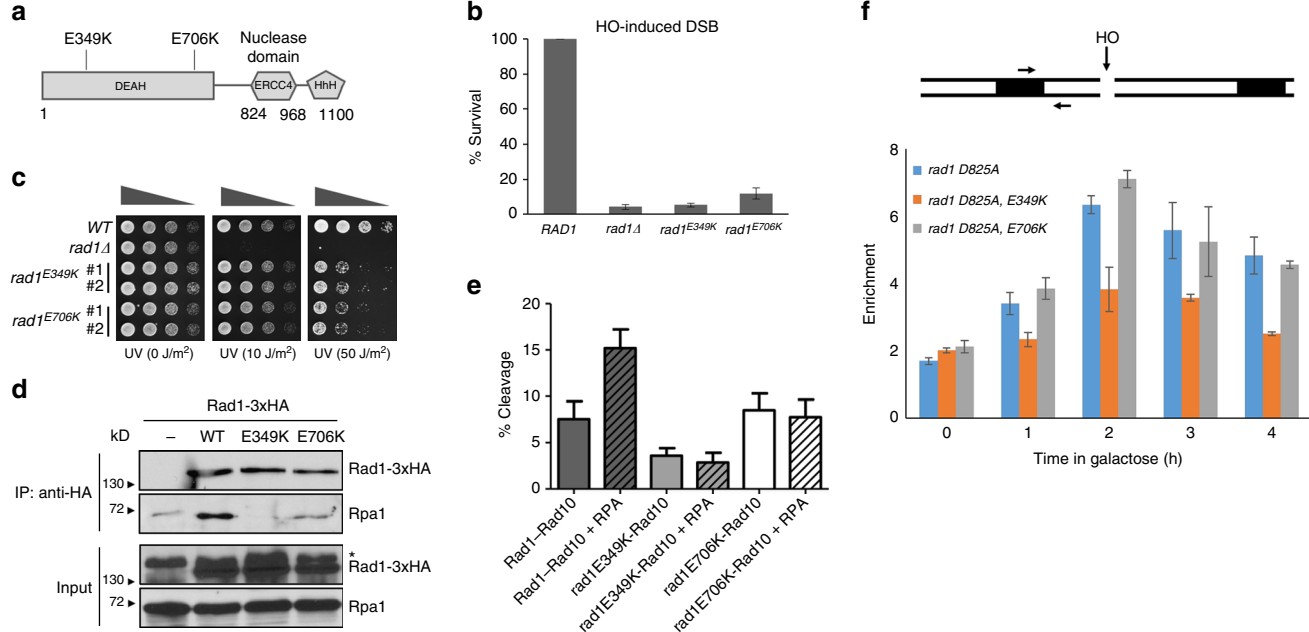

**Fig. 4** *rad1* mutants deficient in recombination and base excision repair (BER). **a** The location of amino acid substitutions in *rad1* mutants deficient in recombination and BER. **b** Percent survival of yeast cells expressing wild-type or mutant rad1 after induction of a double strand break in chromosomally integrated SSA reporter. The results are the average of three independent experiments + s.d. **c** Serial dilutions of indicated yeast strains grown to log phase and treated with indicated UV doses. **d** Co-immunoprecipitation of C-terminal 3xHA tagged Rad1, rad1-E349K, or rad1-E706K with Rpa1. *Non-specific signal. **e** Endonuclease activity of His-Rad1–Rad10 (200 nM) on a 3′ flap substrate is stimulated by the presence of yeast RPA (80 nM). Endonuclease activity of His-rad1E349K-Rad10 and His-rad1E706K-Rad10 is unaffected by RPA. Percent endonuclease activity was determined. Quantification was performed using ImageQuant 5.2. Data represents the mean ± SEM of at least four independent experiments with multiple protein preparations. His-Rad1–Rad10 (dark gray), His-rad1E349K-Rad10 (light gray), and His-rad1E706K-Rad10 (white). **f** (Top) the locations of HO break site, primers (arrow), and homologous sequences (boxes). (Bottom) recruitment of rad1 mutants to 3′ flap site. *rad1-D825A* is a nuclease deficient derivative. Error bars are the average ± s.d. of two independent experiments

To gain insight into the basis of recombination and non-NER deficiency in *rad1* mutants, we analyzed several biochemical properties of rad1-E349K and rad1-E706K mutant proteins. First, we examined the expression level of the two rad1 mutants by immunoblot assay. Both rad1-E349K and rad1-E706K are expressed at a level almost identical to wild-type Rad1 (Supplementary Fig. 7a). We then determined whether the purified mutant rad1–Rad10 complexes were proficient in binding and cleaving a branched DNA substrate[28]. The purification profiles of each mutant complex were similar to that of the wild-type protein complex and we observed no differences in expression level in *E. coli*. The extent of binding of both rad1-E349K/Rad10 and rad1-E706K/Rad10 to a splayed Y DNA substrate was similar to that of wild-type Rad1/Rad10 binding in an electrophoretic mobility shift assay, although both mutant protein complexes bound better to the splayed substrate than wild-type at 100 nM (Supplementary Figs. 8a, 15). Notably, the pattern of the shifts in these assays were distinct, particularly the smeary pattern of rad1E349K-Rad10. Rad1–Rad10 and rad1–Rad10 complexes often exhibit smeary shifts, although rad1E349K-Rad10 is particularly so, consistent with either complexes that are unstable as they migrate through the gel or the formation of multiple complexes. As a result, we quantified all of the shifted material to determine binding activities.

Nonetheless, both mutant complexes were able to cleave splayed Y or 3′ flap DNA substrates at least as efficiently as wild-type Rad1/Rad10 in vitro (Supplementary Figs. 8b, c, 15). rad1E706K-Rad10 cleavage activity was enhanced relative to the wild type in the presence of both substrates. In contrast, the cleavage activity of rad1E349K-Rad10 is more similar to the wild-

type activity. Therefore, our in vitro data indicate that the non-NER and recombination defects observed in strains carrying these mutations cannot be explained simply by defects in biochemical affinity and cleavage toward branched DNA molecules.

Next, we determined the integrity of the interaction between rad1-E349K or rad1-E706K and Rad10, Slx4, or Saw1, which are critical for the removal of 3′ NHTs in recombination[36]. We discovered that rad1-E706K retains a strong interaction with Rad10 and Slx4 in the yeast two-hybrid assay (Supplementary Fig. 7d, e), whereas rad1-E349K showed a mild reduction in interaction with Slx4 (Supplementary Fig. 7e). The reduced interaction between mutant rad1-E349K and Slx4 was also observed by co-immunoprecipitation (co-IP) (Supplementary Fig. 7b). The observation is consistent with a recent study that Slx4 interactions with *Xenopus* Xpf were compromised in the presence of point mutations in this region[56]. In contrast, both rad1 mutants interact with Saw1 at a level indistinguishable from that of wild-type Rad1 (Supplementary Fig. 7c). Similarly, both rad1E349K-Rad10 and rad1E706K-Rad10 co-purified with Saw1 (Supplementary Fig. 7f), as described previously for Rad1–Rad10–Saw1[28], including through a gel filtration step.

Finally, we examined the interaction between rad1 mutants and Rpa1, the large subunit of the single-strand binding replication protein A (RPA) complex. RPA has been implicated in positioning XPF–ERCC1 for 5′ incision and thereby stimulates UV lesion repair[57, 58]. Similarly, we observed that RPA stimulated the 3′ flap cleavage by purified Rad1–Rad10 in vitro (Fig. 4e, Supplementary Figs. 9, 16). In the presence of RPA, the efficiency of Rad1–Rad10 on its sub-optimal 3′ flap substrate (with no gap) improved ~2-fold. Titration of RPA led to increasing cleavage

efficiency[48]. In contrast, titration of *E. coli* single-strand binding protein (SSB) did not stimulate Rad1–Rad10 cleavage activity (Supplementary Figs. 9, 16). This observation indicates that the RPA stimulation of Rad1–Rad10 is specific and that the interaction between Rad1–Rad10 and RPA is important for the stimulation of catalytic activity toward 3′ flap substrate.

Using immunoprecipitation (IP) assays, we discovered that both rad1 mutants were severely defective in their ability to interact with Rpa1 (Fig. 4d, Supplementary Fig. 13). Furthermore, RPA failed to stimulate 3′ flap cleavage by either rad1E349K-Rad10 or rad1E706K-Rad10 in in vitro nuclease activity assays (Fig. 4e, Supplementary Figs. 9, 16). The results suggest that the rad1 mutants are defective in RPA interaction, and the interaction with RPA is critical for non-NER functions but is largely dispensable for UV lesion repair. Importantly, rad1E706K, but not rad1-E349K, is efficiently recruited to 3′ NHT in vivo, as determined in our chromatin immunoprecipitation (ChIP) assay (Fig. 4f). This indicates that the interaction between Rad1 and Rpa1 is likely dispensable for the recognition or the stable association of Rad1–Rad10 to recombination substrates.

**Rad1 mutants sensitize *mus81Δ* cells to ICL agents.** Identification of the two new rad1 mutants that are selectively deficient in recombination and non-NER activity enabled us to assess the role of distinct Rad1–Rad10 activities in replication-coupled ICLR occurring at the S/G2 phase of the cell cycle (i.e., Rad1–Rad10–Saw1 versus Rad1–Rad10–Rad14). Should the non-NER activity be important for replication-coupled ICLR, expression of rad1-E349K or rad1-E706K would lead to HN2 sensitivity in *MUS81*-deleted cells when arrested and released from G1 into S. We found that *rad1-E349K* and *rad1-E706K* each sensitized *mus81Δ* cells to HN2 at S/G2 albeit not as severely as *saw1Δ* (Fig. 5a, b). The *mus81Δ* cells expressing mutant rad1 derivatives also showed hypersensitivity to CDDP compared to *mus81Δ* mutants (Supplementary Fig. 10). The results support the hypothesis that the role of Rad1–Rad10 in replication-coupled ICLR can be uncoupled from its role in UV lesion repair.

**XPF^E239K and XPF^E569K are deficient in ICLR.** Strong sequence conservation between XPF and Rad1 at or near the amino acid residues mutated in our rad1 mutant prompted us to test whether

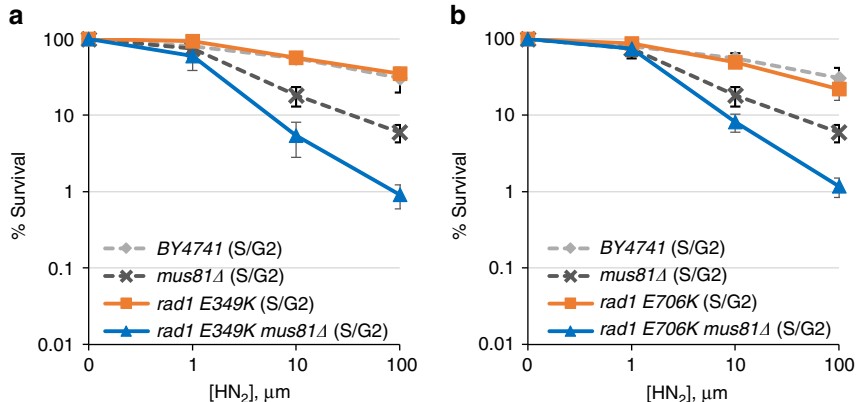

**Fig. 5** Sensitivity to HN2 treatment of cells expressing rad1-E349K (**a**) and rad1-E706K (**b**) mutants and *mus81* deletion. Cells were arrested in G1 by α-factor treatment and released into medium with the indicated concentrations of HN2 before plating. Error bars are average ± s.d. of three independent experiments

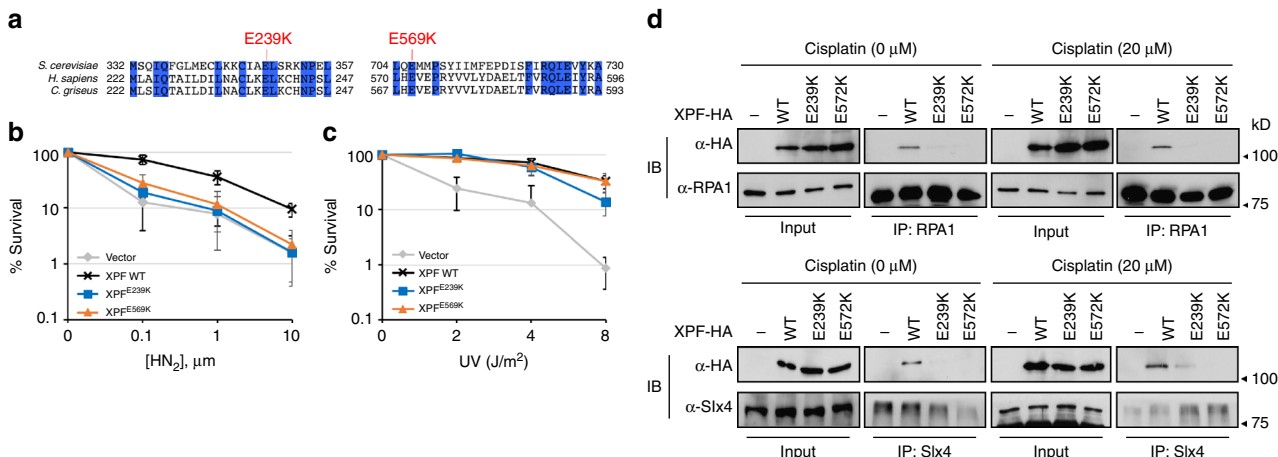

**Fig. 6** Effect of XPF mutations on ICLR. **a** Homology between XPF sequences in *S. cerevisiae, H. sapiens,* and *C. griseus* at the regions flanking *rad1* mutations. The amino acids identical to all three species are shown in blue. **b, c** Percentage of survival of XPF^E239K and XPF^E569K was calculated by dividing the number of exposed cells with the number of non-exposed cells upon treatment of different doses of HN2 (**b**) and UV (**c**). **d** Co-immunoprecipitation of C-terminal HA tagged XPF, XPF^E239K, or XPF^E572K with Rpa1 or SLX4. 293T cells were transfected with pcDNA3.1-XPF-HA or the mutant variants and subjected to 20 μm cisplatin treatment for 2 h. Cell lysates were prepared and analyzed by immunoprecipitation and immunoblot assays with antibodies as described in Methods. Input (2%) is shown

**Table 1 DSB-induced recombination frequencies in UV41-AK45**

| Stably transfected plasmid | TK(-) induced frequency[a] (average ± SEM)×10$^{-2}$ | Fold change |
|---|---|---|
| pcDNA3.1 | 0.53 ± 0.14 | 1.00 |
| pcDNA3.1-XPF | 1.57 ± 0.16 | 2.99 |
| pcDNA3.1-XPF E239K | 0.69 ± 0.18 | 1.30 |
| pcDNA3.1-XPF E569K | 0.64 ± 0.11 | 1.21 |
| pcDNA3.1-XPF L230P | 0.42 ± 0.05 | 0.79 |
| pcDNA3.1-XPF C236R | 0.33 ± 0.07 | 0.62 |

[a]Induced frequencies corrected for background colony formation

XPF mutated at the analogous amino acid residues would also exhibit an ICLR defect in mammalian cells. We constructed shuttle vectors expressing Chinese hamster XPF$^{E239K}$ and XPF$^{E569K}$ (analogous to rad1-E349K and rad1-E706K, respectively, Fig. 6a), and the plasmids expressing the mutant versions of XPF were introduced to the XPF-deficient CHO cell line UV41[46]. The level of mutant XPF expression is indistinguishable from that of wild type (Supplementary Fig. 11). We then examined UV, HN$_2$, and Mitomycin C (MMC) sensitivity in cells expressing mutant or wild-type XPF by survival and colony formation assays. If the branched DNA cleavage activity (analogous to Rad1–Rad10–Saw1 activity in yeast) is critical for mammalian ICLR, cells expressing the XPF mutants should be sensitive to HN$_2$ and MMC, but not to UV. In accordance with our hypothesis, cells expressing either XPF$^{E239K}$ or XPF$^{E569K}$ did not complement the XPF-deficiency of UV41, resulting in sensitivity to HN$_2$ in survival assays using trypan blue dye exclusion staining (Fig. 6b). Cells expressing XPF$^{E239K}$ were also sensitive to HN$_2$ in clonogenic assays (Supplementary Fig. 12b). Puzzlingly, cells expressing XPF$^{E569K}$ were not sensitive to HN$_2$ in clonogenic assays (Supplementary Fig. 12b). Expression of XPF$^{E569K}$ also did not sensitize cells to MMC (Supplementary Fig. 12a). Expression of these two mutants complements UV sensitivity in the UV41 cell line to a level identical (-E569K) or only slightly less (-E239K) to that of wild-type XPF (Fig. 6c, Supplementary Fig. 12c).

Mutations in ERCC1 or XPF were previously shown to reduce SSA between *APRT* gene repeats after I-*Sce*I induction likely due to deficiency in 3′ NHTR[46]. We discovered that expression of XPF$^{E239K}$ or XPF$^{E569K}$ led to a 2–3-fold reduction in TK-events compared to that of XPF, a level indistinguishable from that observed in xpf-deficient UV41 cells (Table 1). These results indicate that, as in yeast, these xpf mutations successfully discriminate between distinct XPF–ERCC1 functions in ICLR; the one involving cleavage of NER type DNA intermediates and the other involving branched DNA molecules.

Lastly, we examined the integrity of interaction between XPF variants tagged with 3xHA and two likely factors involved in replication-coupled ICLR, Slx4, and Rpa1, using IP with anti-HA, Slx4, and Rpa1 antibodies. We detected physical interaction between XPF-3xHA and Rpa1, and confirmed the robust interaction between XPF-3xHA and Slx4 under CDDP or mock treatment conditions (Fig. 6d, Supplementary Fig. 14). Most importantly, both XPF mutants (-E239K and E572K) are defective in interaction with Slx4 and Rpa1. The results suggest that the interactions between XPF, Slx4, and Rpa1 are important for replication-coupled ICLR but dispensable for UV resistance.

**FA causing XPF mutations are deficient in 3′ NHTR in yeast.** Mutations in XPF in humans lead to Xeroderma Pigmentosum (XP), which causes extreme sensitivity to UV light. Furthermore,

xpf mutations also lead to FA or FA-like (-L230P and -C236R)[32, 33] and render patient cells severely sensitive to ICL-inducing agents. Intriguingly, two FA-causing mutations were mapped to the region of XPF (C230P and C236R) that aligns near the yeast Rad1-E349K mutation (Fig. 7a). We created the equivalent mutations in yeast, based on alignment with XPF (Fig. 7a) by integrating *rad1M340P* and *rad1C346R*, equivalent to XPF$^{C239P}$ and XPF$^{C236R}$, respectively, into the endogenous genomic *RAD1* locus and measured the frequency of SSA and gene conversion, both of which require 3′ flap removal. We found that *rad1C346R* exhibited a severe defect in SSA (Fig. 7b). The *rad1M340P* mutation also showed a significant deficiency in SSA. These two mutations, however, display very different UV sensitivity: *rad1M340P* is extremely sensitive to UV, whereas *rad1C346R* is no more sensitive to UV than wild-type cells (Fig. 7c).

We then examined the HN$_2$ and UV sensitivity profiles of hamster cells expressing XPF$^{C230P}$ and XPF$^{C236R}$. Both XPF mutants were sensitive to HN$_2$ treatment; only XPF$^{L230P}$ exhibited a mild sensitivity to UV (Fig. 7d, e). We also determined the frequency of TK-, intra-chromosomal recombination event between *APRT* direct repeats upon induction of I-*Sce*I from cells expressing FA mutants. We discovered that both XPF mutants are deficient in intra-chromosomal recombination yielding TK- cells after I-*Sce*I induction (Table 1). These data indicate that this region in XPF might be critical for recombination and ICLR, but dispensable for NER.

**Discussion**

XPF–ERCC1 (Rad1–Rad10 in yeast) forms the core component of ICL-induced and UV-induced DNA damage repair. Cells deficient in either subunit show severe sensitivity to both UV-inducing and ICL-inducing agents[29, 39]. By analyzing the effect of deleting *SAW1* on ICL sensitivity, we have demonstrated that Rad1–Rad10–Saw1 is important for replication-coupled ICLR and functions at stressed replication forks, likely in a non-NER step. The Rad1–Rad10–Saw1-dependent activity is at least partially redundant with that of another structure-specific endonuclease, Mus81-Mms4. We further identified rad1 mutants that selectively disabled non-NER functions but left UV lesion repair function largely intact in the genetically tractable yeast system. The involvement of XPF–ERCC1 in non-NER steps within ICLR seems conserved across species because the analogous mutations in XPF led to defective replication-coupled ICLR but retained NER activity.

We revealed specific and distinct roles for Rad1–Rad10–Saw1 in replication-coupled (S/G2) and uncoupled (G1) ICLR. Deletion of *RAD1* or *RAD14* that sensitizes cells at all cell cycle stages to both HN$_2$ and CDDP treatments highlights replication-uncoupled ICLR, wherein Rad1–Rad10 presumably contributes to ICL unhooking by incising 5′ to ICLs as it does to UV lesions in NER. Replication-uncoupled ICLR (a.k.a. Replication Independent Repair; RIR) has been described in both yeast and mammals and NER has been implicated in initiation of this repair[4, 5, 7, 47]. Indeed, our data, as well as that from others, show that yeast cells are extremely reliant on NER factors for survival following ICL damage[29], which may indicate that replication-uncoupled ICLR is the major ICLR pathway in yeast. Yeast cells may clear ICLs more rapidly due to the intrinsically high gene density within their genome. Alternatively, ICLs occurring within highly transcribed genes may be rapidly identified and resolved by transcription-coupled NER repair machinery[59].

While deletion of *RAD1* or *RAD14* sensitizes cells to ICL agents at all cell cycle stages, deletion of *SAW1, SLX4*, or expression of rad1 mutants deficient in non-NER functions revealed a distinct ICLR pathway that operates preferentially in S/G2 phase cells and

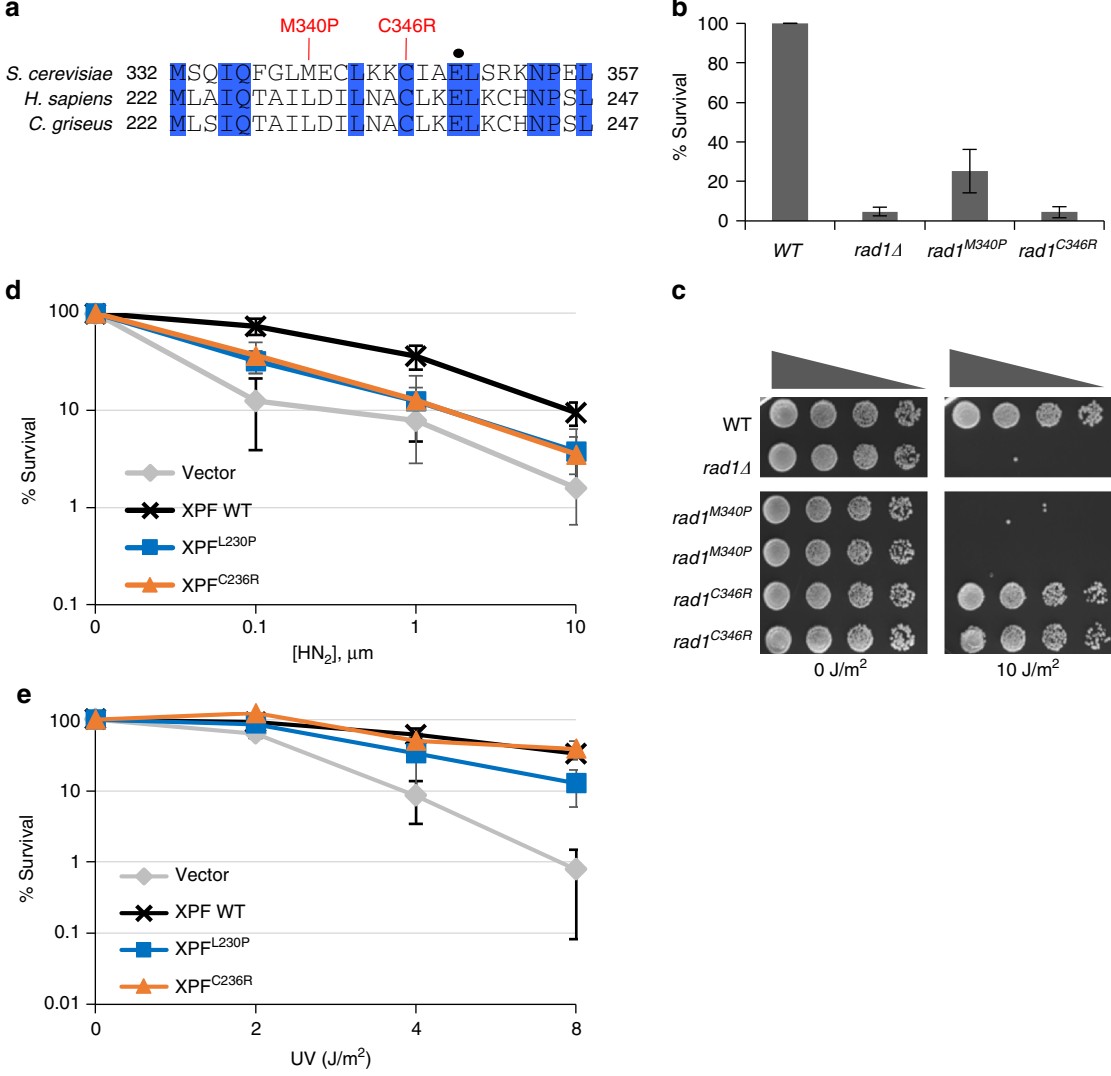

**Fig. 7** XPF mutations that result in Fanconi anemia or combined Fanconi anemia/Cockayne Syndrome are deficient in 3′ flap removal. **a** FA causing XPF mutations are located near XPF-E239. Conserved residues are highlighted in blue. Mutations analogous to human mutations that result in Fanconi anemia or Fanconi anemia/Cockayne Syndrome are indicated. Rad1-E349 is indicated by a black dot. **b** The effect of FA-causing rad1 mutations on single-strand annealing. **c** Serial dilutions of *rad1* mutants analogous to FA-causing XPF mutants after UV (10 J/m$^2$) treatment. **d**, **e** Percentage of survival of FA causing XPF mutants after treatment of different doses of HN$_2$ (**d**) and UV (**e**)

leads to sensitivity to HN$_2$ only when *MUS81* is deleted. The phenotypes of rad1 mutants are significantly milder than a complete loss of *RAD1*, indicating a sub-pathway. We propose that ICLR at S/G2 in yeast is equivalent to replication-coupled ICLR in mammals. The timing of operation (S/G2) and likely components associated with these processes (Slx1, Slx4, and Mus81) support this premise and indicate that a significant part of the process might be conserved across species. Along with its well-defined replication dynamics and biochemical components, yeast offers an exciting opportunity to dissect underlying mechanisms of replication-coupled ICLR common from yeast to human.

Recent evidence suggests that XPF–ERCC1 contributes to ICLR in a manner that is distinct from its function in NER, with the complex catalyzing ICL unhooking steps by nicking stalled replication forks[23, 24, 56, 58]. Alternatively (or perhaps in addition to ICL unhooking), XPF–ERCC1 might participate in ICLR as a 3′ NHTR endonuclease at the post-strand invasion steps during HR[35, 60]. Several lines of evidence suggest that Rad1–Rad10–Saw1 (-Slx4) cleaves stalled replication forks and thereby triggers ICL

unhooking in replication-coupled ICLR. First, Saw1 is a structure-specific DNA-binding protein with high affinity to 3′ flap and replication fork-like DNA substrates[28]. Its unique DNA binding property could, in principle, help target the Rad1–Rad10–Saw1 complex to recombination substrates and stressed replication forks[28, 36], both of which likely constitute key intermediates for replication-coupled ICLR in yeast and human cells. Second, we observed that Saw1 (presumably in complex with Rad1–Rad10) forms nuclear foci in S/G2 cells upon HN$_2$ treatment and co-localizes with Mrc1, which marks stalled replication forks. Using time-lapse microscopy analysis of Saw1 and Rad52 foci formation, we demonstrated that the formation of Saw1 foci is independent of Rad52 and precedes those of Rad52. Furthermore, Saw1- and Rad52 foci did not co-localize in the presence of ICL damage, although they did co-localize in the presence of MMS-induced damage. These results indicate that a temporal gap exists between the activities of Rad1–Rad10–Saw1 and Rad52 in ICLR, and Rad1–Rad10–Saw1 is not merely involved in processing recombination intermediates during ICLR. We propose that Rad1–Rad10–Saw1 recognizes stalled replication forks early in

ICLR and acts at more than one step in ICLR, such as in fork cleavage and the processing of recombination intermediates.

Our results revealed functional redundancy between Rad1–Rad10–Saw1 and Mus81-Mms4 in replication-coupled ICLR. The relationship between Rad1–Rad10–Saw1 and Mus81-Mms4 in ICLR is consistent with previous reports of overlapping functions of XPF–ERCC1 and MUS81-EME1 in resolving late replication intermediates at fragile sites and in HR[61–63]. In yeast, Rad1–Rad10 prefers to cleave 3′ flaps that have a ssDNA gap at the junction, whereas Mus81-Mms4 preferentially cleaves 3′ flaps without a gap[64]. Vertebrate cells also possess FAN1, yet another structure-specific endonuclease capable of cleaving 5′ (upstream) of the 3′ flap intermediate[65]. Therefore, the interplay between nucleases during resolution of stalled replication forks at ICLs could underlie dynamic changes in the structure of stalled replication forks and the distinct substrate specificity of structure-specific nucleases. Alternatively, Mus81-Mms4 (Eme1 in mammals) might process ICL lesions that are left unrepaired when Rad1–Rad10–Saw1 is absent by virtue of its ability to cleave tangled DNA molecules at the late S/G2 phase of the cell cycle, effectively acting as a back-up system. Indeed, evidence suggests that Mus81-induced DNA breakage is the late ICLR event in human cells[3].

We identified two rad1 mutations that were deficient in BER and 3′ NHTR but remained proficient in NER functions. Biochemical and genetic analyses of these rad1 alleles demonstrated that the defect in cleaving branched DNA and 3′ flaps is likely due to compromised interaction between rad1 and the RPA complex. In a reconstituted nuclease reaction, RPA stimulates wild-type Rad1–Rad10 cleavage of 3′ flap DNA substrates. In contrast, the Rad1–Rad10 mutant complexes were refractory to the stimulatory effect of RPA (Fig. 5e, S9). RPA has been implicated in positioning of XPF–ERCC1 to the target DNA molecule and directing the cleavage process in NER[57, 66]. Biochemically, the RPA complex stimulates incision of an ICL at a model replication fork by XPF–ERCC1[58]. Our results suggest that the Rad1–RPA interaction is critical for non-NER function of the nuclease complex. We speculate that without correct positioning of Rad1–Rad10 at the branched DNA, RPA may become inhibitory to the cleavage by Rad1–Rad10 by binding and occluding the nuclease's natural cleavage site. During NER, the formation of a large NER complex may partially offset the inhibitory action of RPA and XPF–ERCC1 might still gain enough access to the target site.

ERCC1 and XPF mutations in patients underlie a wide range of symptoms, many of which cannot be explained solely by the complex's role in UV lesion repair. For instance, severe sensitivity to ICL agents as shown here and the premature aging found in many XPF–ERCC1 patients and mouse models may be a result of defects in non-NER-related functions[67]. It will be important to elucidate the molecular basis of each symptom found in XPF–ERCC1 deficiency and assign them to branched DNA cleavage deficiency, NER defect, or both. Furthermore, XPF–ERCC1 status has been extensively explored as a predictive biomarker for platinum-based drug treatment that induces ICL lesions[68, 69]. Deeper knowledge of the wide range of cellular functions associated with XPF–ERCC1 and its nuclease activities will help target corresponding XPF–ERCC1 activity to improve its utility as a biomarker for treatment outcomes, and to improve the therapeutic index of patients undergoing treatments with anti-neoplastic agents[70].

## Methods

**Strains**. Strains used to perform experiments are listed in Supplementary Table 1. Gene deletions were created by transforming drug-resistant, PCR-derived DNA sequences of the KANMX, CLONAT, HYGROMYCIN B, or BLASTICIDIN

modules or PCR-derived, yeast metabolic marker genes URA3, LEU2, and HIS3 into the corresponding strains. Most microscopy experiments were performed using the clones from the commercially available Yeast GFP Library (Invitrogen) that fuses GFP to the 3′ end of the gene of interest. For Rad52-mRFP construction, the W303 (RAD5+) RAD52-mRFP1 strain was obtained from Dr. Rodney Rothstein[71] and modified by insertion of CLONAT maker distal to the stop codon. This strain was then used as a template to amplify the RAD52-mRFP::CLONAT DNA sequence and was subsequently transformed into the GFP library strains to fluorescently label Rad52 with red fluorescence.

**Screening for mutants deficient in 3′ flap cleavage**. The RAD1-3HA was subcloned into pRS425-ADH1 using BamHI and XhoI restriction enzyme sites and mutagenized by hydroxyamine treatment. The mutagenized plasmid was then transformed into rad1Δ apn1Δ mutant cells. Colonies were patched to leucine-deficient media and incubated at 30 °C overnight. These plates were then replica plated to YEPD with or without 0.011% MMS. For UV treatment, the plates were treated with 60 J/m$^2$ using Stratagene UV cross-linker with the dose rate of 10 J/second. Clones sensitive to MMS were picked and analyzed individually by spotting assay. Plasmid from cells displaying hypersensitivity to MMS but resistance to UV were isolated and transformed into E. coli DH5α. Three to five independent colonies for each individual mutant were isolated, and re-transformed to apn1Δ rad1Δ strain for MMS and UV sensitivity to further confirm their phenotypes. The rad1 mutants showing sensitivity to MMS only were then subjected to plasmid-based 3′ flap removal assay and SSA assay, respectively, to fully verify their 3′ flap cleavage deficiency.

**SSA assay**. EAY1141 and the rad1 gene deletion derivatives bearing empty vector, wild type RAD1, and the rad1 mutant identified from the screen were grown in leucine-deficient media, then switched to YEP-glycerol for 6–8 h before plating to either SC-leucine or YEP-galactose plates. The plates were incubated at 30 °C for 3–4 days. Survival was calculated by dividing total number of viable colonies on YEP-galactose plates by the number of viable colonies on SC-leucine plates.

**Plasmid-based SSA assays**. Log-phase cells were transformed with either Bsu36I-digested or mock-digested pNSU208 by LiOAc transformation protocol. Cells were then plated on media lacking leucine and incubated at 30 °C for 3–4 days. Percent survival was calculated by dividing the number of viable colonies from the Bsu36I-digested transformants by the number of colonies on mock-digested plates as SSA repair frequency.

**Measurement of 3′ flap cleavage efficiency**. pFP120 and pFP122, yeast centromeric plasmids carrying two copies of inverted lacZ sequences, one of which contains an HO recognition site and 610 and 320 bp of sequence deletions[72] were transformed into RAD1 and the rad1 mutant derivatives with the mutant HO recognition site at the MAT locus. Yeast transformants were then cultured in SC media lacking uracil and further incubated in pre-induction YEP-glycerol media for 8 h at 30 °C before plating onto YEP-galactose and YEP-dextrose plate. After 3 days of growth at 30 °C, the colonies growing on YEP-galactose plate were replicated to SC-uracil plate to measure the percentage of plasmid retention as a measure of the efficiency of 3′ flap cleavage by Rad1–Rad10. The percentage of plasmid retention was calculated by dividing the number of colonies growing on SC-uracil plate by the number of cells growing on YEP-galactose.

**DNA damage sensitivity assays**. Exponentially growing BY4741 and its mutant derivatives were subjected to G1 arrest by treating with α-factor (10 μg/ml final) for 2 h at 30 °C. Cells were then washed twice with 10 ml pre-warmed H$_2$O and then re-suspended with pre-warmed 10 ml phosphate buffered saline (PBS) with 10 μg/ml alpha factor (G1) or pre-warmed media for another 20–30 min at 30°C before incubating with HN$_2$, or CDDP at 2 h at 30 °C with rotation, or exposed to 254 nm UV light at the indicated dose. Samples were subsequently plated onto YPD plates and incubated at 30 °C for 3 days before scoring survival.

The percent survival was calculated by dividing the number of viable colonies on drug or UV treated plates by the number of viable colonies from untreated plates and then multiplying this factor by 100. Error bars represent the standard deviation for at least three independent experiments.

**Live-cell imaging assays**. Cell cultures were grown in YEPD overnight and then diluted into fresh YEPD media. After 4 h, 1 ml aliquot was removed as an untreated sample, while the remaining culture was treated with freshly prepared aqueous HN$_2$ at a final concentration of 100 μm. Treated cultures were then incubated at 30 °C incubator in a circular rotator. Aliquots of HN$_2$-treated cultures were removed at indicated time points and mixed on a glass microscope slide with equal volumes of 42 °C media containing 1.5% (w/v) low melting agarose (Agarose Unlimited, Gainesville, Florida) and 100 μm HN$_2$. Images were captured using an Electron Multiplying CCD (EM-CCD) camera mounted to an Olympus IX71 inverted microscope with a 100×, 1.4 NA oil immersion objective. The fluorescent light source was a 250 W Xenon light source. Fluorophores were imaged using the Live Cell Filter Set (Applied Precision, Issaquah, Washington) including mRFP 575/632

nm and GFP 475/525 nm fluorescent filters. Image acquisition times for fluorophore-fusion proteins are as follows: 600 ms for Saw1-GFP and 600 ms for Rad52-mRFP and Mrc1-mRFP. Images were acquired for at least 10 different focal planes at 0.3 μm intervals along the Z-axis of the cells. Images were obtained, deconvoluted, and analyzed using SoftWorx Suite software (Applied Precision, Issaquah, Washington). Contrast enhancement was optimized for quantification of each fluorophore and all images were analyzed with the same optimized parameters. Graphs depict the average of three experiments with at least 150 cells counted for each experiment for at least a total of 450 cells counted per experiment. Error bars represent the standard deviation for at least three independent experiments.

**Yeast two hybrid**. Yeast two hybrid was performed in order to assess the ability of rad1-E349K and rad1-E706K to interact with Rad10 and Slx4. *RAD10* and *SLX4* were cloned into pGADT7 and *RAD1* was cloned into in the pGBKT7 (Invitrogen). The rad1-E349K and rad1-E706K single point mutations were introduced in the pGBKT7-*RAD1* by site-directed mutagenesis. Plasmids were transformed into opposite mating types (a or α) and mated to create diploid strains harboring both plasmids. Interactions were assayed according to manufacturer's protocol.

**Co-immunoprecipitation**. *RAD1*, rad1-E349K, and rad1-E706K were tagged with a C-terminal 3HA tag at their genomic locus. Yeast lysates expressing Rad1-3HA, rad1-E349K-3HA, and rad1-E706K were prepared by lysing cells with glass beads in 0.6 ml cold IP150 solution (20 mM Tris-HCl, pH 8.0, 150 mM NaCl, 0.5% NP-40) supplemented with protease inhibitor cocktail (Roche Life Science). One microgram purified anti-Saw1 polyclonal antibody was added to pre-cleared cell lysate and incubated at 4 °C for 90 min. Protein G Agarose slurry (30 μl) was then added to the lysate, and the mixture was incubated for an additional 30 min at 4 °C. The beads were collected by centrifugation and washed extensively with IP150, and applied to separation by SDS-PAGE and detected by immunoblotting using anti-HA monoclonal antibody (Roche Life Science). To detect interaction between Rad1 variants and Slx4, the plasmids expressing *RAD1*-3HA, rad1-E349K-3HA, and rad1-E706K-3HA were transformed into cells expressing TAP-tagged Slx4. Rad1-Slx4 complexes were immunoprecipitated by using IgG Agarose beads (Thermo Scientific™ Pierce™), subsequently separated by SDS-PAGE and detected by Western blotting using anti-HA monoclonal antibody (Roche, 11666606001, 1:5000 dilution) or anti-Rpa1 polyclonal antibody (from Dr. Steven Brill, 1:1000 dilution).

For co-immunoprecipitation assay between XPF, RPA, and SLX4, 293T cells (purchased from ATCC and grown in DMEM with 10% FBS) were transfected with plasmids expressing HA-XPF or its mutants (E239K or E562K) and subjected to 20 μm cisplatin treatment for 2 h. Cell extracts were prepared with lysis buffer (20 mM Tris-HCl, pH7.2, 100 mM NaCl, 10% Glycerol, 1 mM EDTA, 0.1% NP-40, 1% Triton X-100), followed by IP with protein G-agarose beads (Sigma-Aldrich) and the antibodies specific to RPA1, or SLX4 (anti-RPA70 and anti-SLX4, Santa Cruz Biotechnology, sc-135225), resolved by SDS-PAGE and immunoblotting with anti-HA (Roche,11666606001, 1:5000 dilution), anti-RPA70 (Abcam, ab79398, 1:1000 dilution), and anti-SLX4 (Abcam, ab189591, 1:500 dilution) antibodies.

**Chromatin immunoprecipitation**. Exponentially growing yeast cell cultures were crosslinked with 1% formaldehyde for 30 min at RT and then quenched by 125 mM glycine for 5 min. Cells were lysed with glass bead and then sheared by sonication to generate around 0.5 kb chromatin fragments. Extracts were divided into input and IP samples (1:10 ratio) and then IP samples were incubated with HA-antibody (Roche) for 2 h at 4 °C before incubated with pre-cleaned agarose G beads for 1 hr at 4 °C. Protein bound beads were washed three times before eluted and reversed crosslinking by proteinase K treatment for overnight at 65 °C. Pulled down DNAs were precipitated after phenol extraction and then used for qPCR. Primers are listed in Supplementary Table 2.

Chromatin associated Mrc1 and Saw1 were detected by immunoblotting of immunoprecipitates after ChIP using anti-GFP (Life Science, A11120, 1:2000 dilution) or anti-Saw1 antibodies (Generated[28], 1:1000 dilution).

**Purification of Rad1–Rad10 and rad1–Rad10 complexes**. Purifications were performed using column chromatography[28]. The *rad1* alleles were subcloned into pJAS21[28] to generate plasmids that co-overexpress either 6xHis-rad1-E349K and Rad10 (pEM1) or 6xHis-rad1E706K and Rad10 (pEM2) in *Escherichia coli*. Protein expression was induced with IPTG to a final concentration of 0.5 mM for 7 h. Induced cells were passed through a French press 3 times to lyse and the lysates were cleared by centrifugation at 95,000×g for 1 h. The cleared lysate was loaded onto phosphocellulose (Whatman), followed by Ni-NTA (Qiagen) and SP-Sepharose (GE Lifesciences). Fractions containing His-Rad1–Rad10 or His-Rad1–Rad10 were concentrated (Amicon) and flash frozen as aliquots and stored at −80 °C. RPA was purified by column chromatography[73].

**Purification of Rad1–Rad10–Saw1 and rad1–Rad10–Saw1 complexes**. His-Rad1, His-rad1-E349K, or His-rad1-E706K were co-overexpressed in *E. coli* with untagged Rad10 and untagged Saw1[28]. Purifications were performed using Cobalt column chromatography[28]. All lysates were cleared in an Ultracentrifuge (Beckman

Coulter) at 95,000×g for 1 h. The higher speed and additional time further cleared the sample and the gel filtration elution profiles were slightly changed although a higher molecular weight complex containing His-Rad1, Rad10, and Saw1 were still observed. Following centrifugation, the lysate was loaded onto a Cobalt column. His-Rad1, His-rad1-E349K, or His-rad1-E706K were eluted, along with associated Rad10 and Saw1, with increasing concentrations of imidazole. The eluates from the Cobalt columns were loaded onto a Superose 6 (10/300, GE Life Sciences) gel filtration column and 750 μl fractions were collected. The presence of Rad1 was detected by silver staining 8% SDS-PAGE gels. The presence of Rad10 and Saw1 was detected by Western blot.

**Endonuclease assays**. Splayed (LS1/LS3) and 3′ flap (LS1/LS3/LS16) substrates were end-labeled and assembled by annealing[74]. Reactions (10 μl) were performed in 50 mM Tris-HCl (pH 8.0), 5 mM MgCl₂, 50 mM NaCl, 5 mM DTT with 0.1 pmol (10 nm) end-labeled substrates and the indicated protein concentrations. Reactions were incubated at 37 °C for 1 h. The mixture was incubated at 37 °C for 1 h. The reaction was deproteinized by the addition of 100 mg Proteinase K and 0.1% SDS followed by incubation at 37 °C for an additional 15 min. The resulting DNA products were electrophoresed through 10% native 1x TBE gels at 250 V for 90 min. The gels were dried and exposed to PhosphorImager screen (Molecular Dynamics) and quantified by ImageQuant (GE).

To test the effect of RPA on endonuclease activity, reactions were performed in RPA binding buffer (40 mM HEPES-KOH, pH 7.5, 75 mM KCl, 5 mM MgCl₂, 1 mM DTT, 5% glycerol, and 100 μg/ml BSA). Yeast RPA (80 nm) was pre-incubated with the 3′ flap substrate for 10 min at 30 °C. Rad1–Rad10, rad1E349K-Rad10, or rad1E706K-Rad10 (200 nm) were then added and reaction mixtures were incubated at 30 °C for 1 h. Reactions were deproteinized by the addition of 100 μg Proteinase K and 0.1% SDS. After a 15 min incubation at 30 °C, the reactions were loaded onto 10% native acrylamide gel and electrophoresed at 250 V in 1x TBE, for 45 min. The gels were dried and then exposed to a PhosphoImager screen. Quantification of cleavage products was carried out using ImageQuant (GE).

**ICL sensitivity of CHO cells expressing mutant XPF**. The XPF^E239K and XPF^E569K mutations were introduced into the pcDNA3.1-XPF by PCR based site-directed mutagenesis. A HpyAV restriction enzyme site is destroyed by the XPF^E239K mutation and the XPF^E569K mutation introduces a DraIII restriction enzyme site for screening positively mutagenized constructs. The identities of the mutations were further confirmed by sequencing.

UV41 that is hemizygous for XPF and CHO-AA8 cells (a gift from Dr. Rodney Nairn) were grown in DMEM-F12 supplemented with 10% FBS and 1% penicillin/streptomycin. Transfection of CHO cells was performed with Lipofectamine 2000 according to the manufacturer's protocol. Briefly, cells were seeded at $5 \times 10^5$ cells/well and the DNA:Lipofectamine mixture (400 μl) was then added to the cells and incubated at 37 °C overnight. Twelve to sixteen hours post-transfection, cells were trypsinized, divided onto a new 6-well plate, and incubated for an additional 24 h prior to drug sensitivity survival assays.

To test drug sensitivity, cells were washed with PBS and appropriate concentrations of freshly prepared ICL drug were added. Following 2 h treatments at 37 °C, cells were washed one time with PBS and the fresh DMEM-F12 supplemented with 10% FBS was added to incubate for 48–36 h at 37 °C. Cells were then trypsinized and the number of viable cells was counted after the addition of Trypan Blue.

**Uncropped images**. Uncropped images of all blots in the main manuscript can be found in Supplementary Figs. 13-16.

**Data availability**. The data that support the findings of this study are available from the corresponding author upon request.

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

## Acknowledgements

We thank J. Haber, R. Kolodner, R. Nairn, R. Rothstein, and X. Zhao for reagents and the plasmids for the study. We thank Dr. Mark Sutton for providing *E. coli* SSB and for helpful discussions. We are also grateful to the members of the Lee and Surtees labs for helpful discussions and to Dr. Eugen Minca for performing some initial experiments. This work was supported by William and Ella Owen Medical Research Foundation and NIH research grant GM71011 to S.E.L., ThriveWell Cancer Foundation to E.Y. S., ES022054 and CA188032 to P.H., GM097177 to I.J.F., IMSD R25 GM095459, and a diversity supplement to GM066094 to M.M.R. and GM87459 to J.A.S.

## Author contributions

Conceptualization: F.L., C.H., E.Y.S., J.A.S. and S.E.L.; investigation: J.H.S., C.H., X.L., C. K., F.L., R.E. and M.M.R.; resources: I.F.G., I.J.F. and P.H.; writing—review and editing: J. H.S., C.H., C.K., F.L., E.Y.S., J.A.S. and S.E.L.

## Additional information

**Competing interests:** The authors declare no competing interests.

