## [Peer Review File · Nature Communications]

Reviewers' comments:

Reviewer #1 (Remarks to the Author):

Interstrand crosslink repair (ICLR) is a complex repair pathway that makes use of many genes involved in a variety of other pathways and can occur in replication-dependent and -independent manner. Human and yeast cells have been frequently used to study ICLR and although many of the same genes are involved a key difference is the role of how the unhooking step (initial separation of the two crosslinked strands) is carried out in replication-dependent ICLR. In yeast, this incision depends on the nucleotide excision repair pathway, while in mammals, unhooking depends on the 5' NER endonuclease ERCC1-XPF (Rad10-Rad1 in yeast), but not other NER proteins.

The starting point for this manuscript is the study of how Rad10-Rad1 might contribute to ICL repair outside of replication in yeast. The authors previously showed that Rad10-Rad1 interacts with the Saw1 protein in the ssDNA annealing pathway of homologous recombination and test here is this interaction is required for ICL repair. Here they show that:

1. saw1 Δ cells are indeed sensitive to ICL forming agents together with a deletion in mus81 in S/G2 cells, although not nearly at the level of a deletion of rad1 or other NER proteins. A mutation in saw1 that abolishes the interaction with rad1 leads to similar sensitivity, showing that Rad10-Rad1 and Saw1 act together in this pathway.
2. Saw1 forms foci after treatment with ICL forming agent that are distinct from Rad52 foci after the same treatment, showing that Saw1 foci does not simply contribute to homologous recombination in ICL repair.
3. Mutations isolated in a genetic screen – Rad1-E349K and Rad1-E706K – (i) render cells sensitive to MMS and nitrogen mustard (HN2); (ii) have no defect in interacting with Saw1, Slx4, Rad1 or DNA substrates, yet are deficient in processing flaps in ssDNA annealing, (iii) importantly – have an interaction defect with Rpa.
4. Analogous mutations in human XPF render cells sensitive to crosslinking agents, but not UV.
5. Mutations in XPF that lead to the ICL-sensitive disorder Fanconi anemia, if introduced into Rad1, also cause flap endonuclease activity.

The two main take home messages for me were that a minor NER-independent ICLR pathway exist in S-phase in yeast and that mutations in an interaction between XPF and RPA is conserved from yeast to mammals and is needed for ICL repair and ssDNA annealing activity. These are two important and novel observation worthy of a high-impact publication.

There are however two major issues with the manuscript: the first one is that the organization and writing of the manuscript is very confusing, likely as the story has evolved through many iterations, so that the question that is being addressed and conclusions reached are hidden. The second one is that the biochemical studies in the manuscript are of insufficient quality. Finally, the entire manuscript it in urgent need of thorough language and organizational editing. Publication in Nat Comm therefore seems premature, but addressing the following points should help to improve the manuscript:

Major points:

A. The introduction is written in a very confusing way and does not set up the paper well at all. The principal difference between ICL repair in yeast and humans is unhooking in S-phase dependent ICL repair is NER-dependent in yeast, and NER-independent in humans. ERCC1-XPF is

key for unhooking in humans, and the authors uncover here that Rad10-Rad1 has an NER-independent role in yeast ICL repair as well. They further suggest that a conserved interaction between RPA and XPF/Rad1 is required for ICL repair. The introduction needs to be completely rewritten to clarify this. Specific points:

1. p3. Line 5: It is crucial to differentiate the role of NER in replication-dependent and independent repair ICL repair. NER is not involved in unhooking in replication-dependent ICL repair in higher eukaryotes and the references cited refer to a replication-independent pathway.

2. P3. Line 10: two newer references that clearly distinguish G1 and S-phase ICL repair could be cited here: Muniandy et al, JBC, 2009, 284, 27908 and Williams et al, Mol Cell, 2012, 47, 140.

3. P3. Line 18: It is important to point out that the FA pathway operates in conjunction with replication and that it enables unhooking by ERCC1-XPF. It would also be good to emphasize that the FA pathway is absent in yeast.

4. P3. Line 21: A better reference for the role of ERCC1-XPF in NER would be Sijbers et al, Cell, 1996, 86, 811.

5. P4. Line 9. This is a place where it should be discussed that ERCC1-XPF interacts with SLX4 in higher eukaryotes to mediate ICL unhooking. The following papers should be cited: Klein Douwel et al, Mol Cell, 2014, 54, 460-471, Hodskinson et al, Mol Cell, 2014, 54, 472-484 (current Refs 61, 62).

6. P4. Line 14: It is unclear how similar the ICLR and NER intermediates that are processed by NER are, a more cautionary statement than "Based on structural similarity between the early ICLR intermediate and that of NER" would be appropriate.

7. P4. Line 19: "By contrast" would be better suited than "Surprisingly". The fact that XPA cells are not hypersensitive toward ICL agent has been known for decades.

8. P5. Line 1. Current Refs 59 and 60, Kashiyama et al and Bogliolo et al need to be cited here. The fact that XPF mutations were found that cause FA and not XP is an important piece of information in the understanding of ICLR in humans.

9. P5. Line 17. The authors have not identified mutations in RAD1 that cause a specific defect in BER. This statement needs to be removed here and throughout the manuscript.

B. The biochemical data in Figs 4, 6, S7, S8 are insufficient to support the conclusions reached by the authors. The following specific issues need to be addressed:

10. In many of the figures (S7, S8, 4E), the authors use a Rad1-Rad10 concentration of 200nM with 10nM substrate. Because of the extremely high excess of enzyme over substrate, these are not conditions that allow any tenable conclusions to be made. It is known that Rad1-Rad10 and ERCC1-XPF greatly prefer splayed arm over 3' flap substrates, so appropriate condition for splayed arm cleavage need to be established first. Subsequent studies may then compare how RPA stimulates the cleavage of 3' flap substrates. A recent paper by Abdullah et al (EMBO J. 2017, 36, 2047) provides an example of how this may be done.

11. Fig S8; p.13: It is unclear whether this figure represents cleavage or binding data!! It is essential that the primary data of the gels that give rise to the graphs S8A and S8B is included in the manuscript so that the reader can evaluate the quality of the data. Furthermore, a line graph is much better suited to represent this kind of data than a column graph.

12. Fig 4E: The data in Fig 4E is entirely insufficient. Primary gel data is needed and a time and/or concentration-dependence of the % cleavage activity needs to be shown to show that the various mutations affect cleavage activity in dependence of RPA.

13. The IP data in Figs 6D are skewed by the higher level of XPF-WT in the input lane vs the E293K and E706K mutants. This may artificially lead to a higher signal for interaction with WT vs mutant proteins. This experiment should be repeated with equal protein amounts.

Minor points:

14. p7, Line 22f: Rad1-Rad10-Saw1 contribute to ICLR in S/G2, but independently of "regular" HR, as the authors data with Rad52-deficient cells show. This sentence should reflect that.

15. p9, Line 15; Please format references.

16. p9. Line 18: Expression levels of untagged Saw1 should be shown as supplementary data. This is important information to establish the validity of the data.

17. Fig S7A-C: The legend above the gel needs to be reformatted.

18. p14. Line 6: Two additional papers need to be referenced and mentioned here that show more convincingly how RPA stimulates ERCC1-XPF: For NER: De Laat et al, Genes Dev, 1998, 12, 2598; For ICLR: Addullah et al, EMBO J. 2017, 36, 2047.

Reviewer #2 (Remarks to the Author):

'Distinct roles of XPF/ERCC1 and Rad1/Rad10-Saw1 in replication-coupled and uncoupled inter-strand crosslink repair

In this study Seol et al investigated the mechanism by which Rad1/10 catalyzes ICL repair. To this end, the authors examined the sensitivity/survival of yeast strains lacking Saw1 and Slx4, which bind Rad1/10 and replication forks (RFs), to ICLs. They discovered that Saw1 and Slx4 (as well as Slx1) are important for replication-coupled ICL repair -measured by decreased survival- in the absence of another structure-specific endonuclease, Mus81. The authors went on to identify RAD1 mutants that disrupt interactions with RPA and impair non-NER functions, without impairing UV lesion repair. They propose that Rad1/Rad10 makes separable contributions to ICL repair depending on the stage of the cell cycle. In G1, Rad1/Rad10 removes ICLs via NER, while later, during S-phase, it facilitates replication-coupled ICL removal, redundantly with Mus81/mms4. The authors also propose that interactions between RPA and Rad1/XPF are critical for ICL repair and recombination in yeast and humans.

Overall, I find that the work provides several interesting/important observations. A highlight is the identification of new mutants in Rad1/XPF that appear to specifically disrupt interactions with RPA and show defects in ICL repair and recombination. In my opinion there are, however, several issues that should be clarified by the authors.

Aspects to consider:

1- It is unclear to me how can the experimental setup used in Figures 1 and S1 discriminate between G1 and S-phase DNA repair. How do the authors know that repair of lesions induced by HN2 or CDDP happens within the 2 hours that cells stay arrested in G1? I would rather expect that even the cells that are treated in G1 (alpha-factor) will repair most of the DNA damage after entry into S-phase, when plated after the 2-hour incubation. Can the authors explain better the logic of this experiment? Monitor DNA repair/DNA lesions during the cell cycle? In addition, do the authors know whether the G1 arrest is maintained after 2+2 hours in alpha-factor? I am not convinced that the current experimental setting is sufficient to distinguish replication-coupled from replication-independent ICL repair.

2- Can the authors explain why does ICL induction in G1 lead to a significant reduction in cell viability when compared to ICL induction in S-phase (in the WT strain)? Compare the BY4741 curve in Figure 1A and S1B.

3- Figure 2B: can the authors include the transmission light image, so that the cell contour is made visible? It is otherwise impossible for the reader to see in which cell cycle stage cells are. It would be good to include DAPI staining of DNA.

4- In Figure 3, how come is Mrc1-GFP nor IPed at t=0 and 1? It is clearly present in the input material. Has this experiment been repeated?

5- Figure 6: In the main text, the authors describe treatment with MMC (page 16, line 23), but in the Figure 6D the label says Cisplatin. Which is it?

6- Figure 6D: The input and IPed XPF-HA-WT is significantly higher than for the E293K and E706K mutants. In my view, this is enough to explain the apparent reduction in RPA1 that is Co-IPed by the mutants. The authors need to address this point in order to ensure that RPA binding is indeed defective.

The anti-SLX4 blots in the IP panel are useless. How do the authors know that the signal comes from SLX4?

7- The authors very often refer to data not shown (5 times). It is unclear why the data is not included. There is plenty of space in the Supplementary Figures.

8- Statistical analysis of the data is missing in all Figures.

9-Figure S7: the +- signals in panels A, B and C are all over the place.

10- Manuscript text: I found the manuscript very difficult to read. The introduction provides a lot of information on the topic, but does not help the reader in understanding the scientific questions addressed with the work performed. The discussion is too long, repetitive and lacks focus. I found it difficult to get a clear picture of what is new with the work. Which are the key findings? Most figure legends are too basic and miss important information for the reader to understand what is displayed in the respective Figure. The abstract should also be improved.

Reviewer #1 (Remarks to the Author):

There are however two major issues with the manuscript: the first one is that the organization and writing of the manuscript is very confusing, likely as the story has evolved through many iterations, so that the question that is being addressed and conclusions reached are hidden. The second one is that the biochemical studies in the manuscript are of insufficient quality.

Finally, the entire manuscript is in urgent need of thorough language and organizational editing.

We revised the manuscript substantially to improve its language and the organization.

Major points:

A. The introduction is written in a very confusing way and does not set up the paper well at all. The principal difference between ICL repair in yeast and humans is unhooking in S-phase dependent ICL repair is NER-dependent in yeast, and NER-independent in humans. ERCC1-XPF is key for unhooking in humans, and the authors uncover here that Rad10-Rad1 has an NER-independent role in yeast ICL repair as well. They further suggest that a conserved interaction between RPA and XPF/Rad1 is required for ICL repair. The introduction needs to be completely rewritten to clarify this. Specific points:

1. p3. Line 5: It is crucial to differentiate the role of NER in replication-dependent and independent repair ICL repair. NER is not involved in unhooking in replication-dependent ICL repair in higher eukaryotes and the references cited refer to a replication-independent pathway.

We added the sentence that NER is not involved in unhooking ICL during replication coupled ICLR in higher eukaryotes.

2. P3. Line 10: two newer references that clearly distinguish G1 and S-phase ICL repair could be cited here: Muniandy et al, JBC, 2009, 284, 27908 and Williams et al, Mol Cell, 2012, 47, 140.

We added these two citations in the revision.

3. P3. Line 18: It is important to point out that the FA pathway operates in conjunction with replication and that it enables unhooking by ERCC1-XPF. It would also be good to emphasize that the FA pathway is absent in yeast.

We added the sentence in the revision that FA pathway enables ICL unhooking by ERCC1-XPF in replication coupled ICLR. We also include the sentence that FA pathway is absent in yeast albeit several components are still conserved.

4. P3. Line 21: A better reference for the role of ERCC1-XPF in NER would be Sijbers et al, Cell, 1996, 86, 811.

We added the citation in the revision.

5. P4. Line 9. This is a place where it should be discussed that ERCC1-XPF interacts with SLX4 in higher eukaryotes to mediate ICL unhooking. The following papers should be cited: Klein Douwel et al, Mol Cell, 2014, 54, 460-471, Hodskinson et al, Mol Cell, 2014, 54, 472-484 (current Refs 61, 62).

We modified the sentence as the reviewer suggested. We also cited two papers in the revision suggested by the reviewer.

6. P4. Line 14: It is unclear how similar the ICLR and NER intermediates that are processed by NER are, a more cautionary statement than “Based on structural similarity between the early ICLR intermediate and that of NER” would be appropriate.

We modified the sentence to “Based on structural similarity (i.e. the presence of ss/dsDNA junctions) between the early ICLR intermediate and that of NER”.

7. P4. Line 19: “By contrast” would be better suited than “Surprisingly”. The fact that XPA cells are not hypersensitive toward ICL agent has been known for decades.

We modified the sentence as the reviewer suggested.

8. P5. Line 1. Current Refs 59 and 60, Kashiyama et al and Bogliolo et al need to be cited here. The fact that XPF mutations were found that cause FA and not XP is an important piece of information in the understanding of ICLR in humans.

We added these citations to highlight the points raised by the reviewer.

9. P5. Line 17. The authors have not identified mutations in RAD1 that cause a specific defect in BER. This statement needs to be removed here and throughout the manuscript.

We added the citation in the revision.

10. In many of the figures (S7, S8, 4E), the authors use a Rad1-Rad10 concentration of 200nM with 10nM substrate. Because of the extremely high excess of enzyme over substrate, these are not conditions that allow any tenable conclusions to be made. It is known that Rad1-Rad10 and ERCC1-XPF greatly prefer splayed arm over 3' flap substrates, so appropriate condition for splayed arm cleavage need to be established first. Subsequent studies may then compare how RPA stimulates the cleavage of 3' flap substrates. A recent paper by Abdullah et al (EMBO J. 2017, 36, 2047) provides an example of how this may be done.

We previously demonstrated that the endonuclease assay conditions used here lead to more efficient cleavage of splayed Y substrates compared to 3' flap substrates (Li et al, 2013 *EMBO J.*) The data from that publication are shown here. Please also see the revised Fig. S8, which shows titrations performed under these conditions with the two different DNA substrates.

In the literature, different groups have examined Rad1-Rad10 activity at both near stoichiometric ratios (e.g. Bastin-Shanower, 2003 *MCB*) and under conditions in which the protein is in excess of the DNA substrate (e.g. Davies et al, 2005 *JBC*), as we have done here. Similar trends are observed. We have performed cleavage assays at a variety of DNA substrate concentrations and have observed the same trends – there is a preference for splayed Y substrate over flap regardless of substrate concentration, but the overall cleavage efficiency is reduced as the DNA substrate concentration is increased (see representative experiments below).

We have chosen to use lower DNA substrate concentrations in order to compare the biochemical activities of the Rad1 complexes – when protein is in excess of DNA, the binding affinity can be approximated at 50% binding activity and these concentrations can be compared between the wild-type and mutant complexes, although we hesitate to report an actual K_d due to the sometimes smeary nature of the gel shifts.

Furthermore, the low DNA concentrations are in fact near stoichiometric with RPA in experiments that examine the ability of RPA to stimulate cleavage. We have done titrations at 2 different DNA concentrations under these conditions – 10 and 20 nM – and have observed the same stimulation.

11. Fig S8; p.13: It is unclear whether this figure represents cleavage or binding data!! It is essential that the primary data of the gels that give rise to the graphs S8A and S8B is included in the manuscript so that the reader can evaluate the quality of the data. Furthermore, a line graph is much better suited to represent this kind of data than a column graph.

We apologize – the binding data was inadvertently omitted from the submitted version of the manuscript. We have included binding line graphs of Rad1-Rad10, rad1E349K-Rad10 and rad1E706K-Rad10, as well as representative gel shift gels, for binding to splayed Y substrates.

We have also presented the endonuclease data as line graphs instead of columns and have added representative endonuclease gels to the figure.

12. Fig 4E: The data in Fig 4E is entirely insufficient. Primary gel data is needed and a time and/or concentration-dependence of the % cleavage activity needs to be shown to show that the various mutations affect cleavage activity in dependence of RPA.

We have performed titrations of RPA to demonstrate stimulation of Rad1-Rad10 activity. These will be published in a separate manuscript that is currently in revision at *Nucleic Acids Research*. The relevant figure is shown below and is cited as such in the manuscript. We note that the addition of RPA does lead to a more prominent lower cleavage product in the presence of Rad1-Rad10. We also observe this product in the absence of RPA. We chose the protein concentration under conditions at which we observed maximal cleavage in the presence of wild-type Rad1-Rad10 to test the effect of RPA on the mutant complexes, which is shown in Fig. S9A, along with RPA only controls.

13. The IP data in Figs 6D are skewed by the higher level of XPF-WT in the input lane vs the E293K and E706K mutants. This may artificially lead to a higher signal for interaction with WT vs mutant proteins. This experiment should be repeated with equal protein amounts.

We re-did the IP experiment by loading the equivalent amount of XPF-WT and those of mutants in the gel. The results should address the reviewer's concern and fully support that mutant XPF proteins are deficient in interaction with RPA1 and Slx4.

Minor points:

14. p7, Line 22f: Rad1-Rad10-Saw1 contribute to ICLR in S/G2, but independently of “regular” HR, as the authors data with Rad52-deficient cells show. This sentence should reflect that.

We did not modify the sentence because the current organization of the manuscript describes the results indicating the HR independent role of Rad1/Rad10-Saw1 in ICLR later. We are concerned that changing the sentence would confuse the reader.

15. p9, Line 15; Please format references.

We formatted the references.

16. p9. Line 18: Expression levels of untagged Saw1 should be shown as supplementary data. This is important information to establish the validity of the data.

We now included the immunoblot assay showing the untagged and GFP tagged Saw1 to ensure their equivalent levels of expression in cells.

17. Fig S7A-C: The legend above the gel needs to be reformatted.

The legend is re-formatted as suggested by the reviewer.

18. p14. Line 6: Two additional papers need to be referenced and mentioned here that show more convincingly how RPA stimulates ERCC1-XPF: For NER: De Laat et al, Genes Dev, 1998, 12, 2598; For ICLR: Addullah et al, EMBO J. 2017, 36, 2047.

These citations are included in the revision.

Reviewer #2 (Remarks to the Author):

1- It is unclear to me how can the experimental setup used in Figures 1 and S1 discriminate between G1 and S-phase DNA repair. How do the authors know that repair of lesions induced by HN2 or CDDP happens within the 2 hours that cells stay arrested in G1? I would rather expect that even the cells that are treated in G1 (alpha-factor) will repair most of the DNA damage after entry into S-phase, when plated after the 2-hour incubation. Can the authors explain better the logic of this experiment? Monitor DNA repair/DNA lesions during the cell cycle? In addition, do the authors know whether the G1 arrest is maintained after 2+2 hours in alpha-factor? I am not convinced that the current experimental setting is sufficient to distinguish replication-coupled from replication-independent ICL repair.

Previous work has demonstrated that similar treatment of cells with ICL agents in G1 results in residual cross-links through the cell cycle; ICL were incised partially immediately after induction of cross-links, i.e. prior to entry into S phase (Grossman et al, 2001). Therefore there is likely a combination of replication-independent and replication-coupled ICLR occurring under these conditions. Nonetheless, the fact that cells arrested and held at G1 after DNA damage exhibited different viability than those released to S support the premise that many G1 arrested cells repair ICL at G1 and not S. Should G1 arrested cells repair ICL ONLY during S/G2 after

the prolonged arrest, the outcomes should be indistinguishable to those released to S/G2 after ICL agent treatment. We also monitored cell cycle profile in each experiment and confirmed that cells were remained arrested at G1 during the entire alpha factor arrest (see Fig. S1B).

2- Can the authors explain why does ICL induction in G1 lead to a significant reduction in cell viability when compared to ICL induction in S-phase (in the WT strain)? Compare the BY4741 curve in Figure 1A and S1B.

The precise reason for the low viability to ICL in G1 in yeast cells is not known. However, it is tempting to speculate that ICLR at S/G2 significantly improves the viability of cells upon ICL damage as compared to that operate in G1. Indeed, deletion of *RAD52* in cells released to S phase reduced the viability slightly above that in G1 cells. We would also like to emphasize that our results are consistent with that reported by McHugh's group in 2005 (see Fig. 3, Mol. Cell. Biol. 2005 vol. 25:2297-2309).

3- Figure 2B: can the authors include the transmission light image, so that the cell contour is made visible? It is otherwise impossible for the reader to see in which cell cycle stage cells are. It would be good to include DAPI staining of DNA.

We have included the transmission light (DIC) image in the figure as requested by the reviewer.

4- In Figure 3, how come is Mrc1-GFP nor IPed at t=0 and 1? It is clearly present in the input material. Has this experiment been repeated?

The results illustrate the levels of Mrc1 pulled down by chromatin immunoprecipitation after DNA damage. Since most Mrc1 proteins do not associate with chromatin without exogenous DNA damage or at cell cycle stage outside of S phase, the level of Mrc1 in immunoprecipitate is almost undetectable at t=0 and only barely detectable at 1 h post damage. The low level at T=1 could also be attributed to the low-resolution limit of the assay.

5- Figure 6: In the main text, the authors describe treatment with MMC (page 16, line 23), but in the Figure 6D the label says Cisplatin. Which is it?

We fixed this error. We apologize for this mistake.

6- Figure 6D: The input and IPed XPF-HA-WT is significantly higher than for the E293K and E706K mutants. In my view, this is enough to explain the apparent reduction in RPA1 that is Co-IPed by the mutants. The authors need to address this point in order to ensure that RPA binding is indeed defective.

The anti-SLX4 blots in the IP panel are useless. How do the authors know that the signal comes from SLX4?

We re-did these experiments and the new results showing the equivalent amount of XPF and the mutant proteins are now included. We also modulated the amount of exposure time in Western blot with anti-SLX4 antibodies to improve the resolution of the Slx4 signal in the assay.

7- The authors very often refer to data not shown (5 times). It is unclear why the data is not included. There is plenty of space in the Supplementary Figures.

We now included and show the results listed as “data not shown” in the original submission as the part of supplementary information.

8- Statistical analysis of the data is missing in all Figures.

We performed statistical analysis of the data where necessary and included in the revision as requested by the reviewer.

9-Figure S7: the +- signals in panels A, B and C are all over the place.

We fixed this issue.

10- Manuscript text: I found the manuscript very difficult to read. The introduction provides a lot of information on the topic, but does not help the reader in understanding the scientific questions addressed with the work performed. The discussion is too long, repetitive and lacks focus. I found it difficult to get a clear picture of what is new with the work. Which are the key findings? Most figure legends are too basic and miss important information for the reader to understand what is displayed in the respective Figure. The abstract should also be improved.

We have revised the entire manuscript to improve the readability and to focus on the key points of the paper. In particular, both the introduction and discussion have been edited and re-focused on the questions at hand, as requested by both reviewers.

Reviewers' comments:

Reviewer #1 (Remarks to the Author):

The revised version of the manuscript is significantly improved over the original submission. The manuscript has been completely worked over. It is now clearly articulated and the two main take-home points - the existence of an NER-independent ICL repair pathway in yeast and the isolation of separation of function mutations that selectively abolish the ICL repair and ssDNA annealing functions of Rad1-Rad10 without affecting the repair of UV lesions by NER - are clearly described. The authors also included additional experiments to solidify the claim that the isolated mutant alleles of Rad1 have a defect in the interaction with RPA and SLX4 and address various additional concerns raised in the review.

Despite these improvements, the biochemical data in the manuscript (Fig 4E, S8, S9) is still of insufficient quality for publication. The figure added from another manuscript cited as NAR, under revision (in response to point 12, reviewer 1) does little to address this concern. In looking at the gel provided, I see no additional product formation with increased RPA concentration. The claim that RPA stimulates incision of WT, but not E349K and E706K mutants is simply not convincingly supported by the experiments.

Perhaps it would be best to remove the biochemical data from the manuscript (as this issue seems to be addressed now in a separate paper anyway), and the paper can then be re-reviewed if it is still appropriate for Nat Comm.

Additional points:

p5. Lines 7ff. While it possible that ERCC1-XPF has multiple roles in ICL repair, only one of them has been specifically defined and this should be clarified here. Refs 69 Klein -Duwel et al and 71, Abdullah et al, clearly describe a role of ERCC1-XPF in the incision point in ICL repair, and these papers need to be discussed here in the introduction, not just the discussion.

p.14. line 2ff and Fig S8A. I strongly disagree that the "binding of E349K and E706K binding to Y DNA was indistinguishable from WT protein: The binding of the three protein shows very distinct patterns of bands, indicating a different binding mode and/or geometry. Furthermore, I disagree with the assessment that endonuclease efficiency is the same. The values appear different at least for the flap substrate and the values in x-axis of the quantification for S8B and S8C do not represent the values shown in the gel. Fig S8 is simply not publication quality and does not support the claims made by the authors.

p.15, line 2: It is not permissible to cite a paper "in revision" - Reference 72 and the claim associated with it needs to be deleted or the data shown in the supplementary material. See my separate comment on the figure in question above.

Reviewer #2 (Remarks to the Author):

Overall, the authors addressed the issues raised and significantly improved the manuscript. In my opinion, the manuscript is almost ready for publication.

Three specific issues that I think should still be addressed are:

Points 1 and 2 - In my opinion, the author's explanation for the viability differences between treatment of G1 or S/G2 cells with ICL-inducing drugs is insufficient. Given that a key conclusion about the relevance of cell cycle stage for ICLR pathway usage is drawn from these genetic experiments, it is important to understand exactly what these experiments reveal. Given that the

WT strain is significantly more sensitive to ICL-inducing drugs in G1 than in S/G2, making comparisons about the requirement of certain pathways for repair becomes very complex. At minimum, the authors should choose a range of concentrations of HN2 that causes comparable levels of cell death in G1 and S/G2, in wild-type cells. Otherwise the plots can be rather misleading.

Point 3- The staging of cells in the images shown in Figure 2 is, in my opinion, highly subjective. The top DIC image (HN2) in panel B shows a cell that can easily be in mitosis. The lower DIC panel (MMS) shows what could be two independent cells. As suggested before, simple DNA staining with DAPI would have helped solving these staging issues.

Point 4- In the main text, line 20, the authors write the following: "Saw1 is present in chromatin fractions pulled down by Mrc1-GFP after HN2 treatment...". In my view, this suggests that what the authors did was to IP Mrc1-GFP in conditions that preserve binding to chromatin, and detect Saw1 in the IPed material. Unfortunately, this is all the information the reader gets to figure out how the experiment was been done. There is also no information in the methods section, except citation of a previous paper, which also cites a previous paper).

In the response to my previous request for clarification of this experiment, in particular to the absence of Mrc1-GFP signal in some of the IPs, the authors write that "the results illustrate the levels of Mrc1 pulled down by chromatin immunoprecipitation. Since most Mrc1 proteins do not associate with chromatin without exogenous DNA damage...the level of Mrc1 in immunoprecipitates is almost undetectable...".

If I understand the authors correctly, what they did was to prepare a chromatin fraction, from which they IPed Mrc1-GFP, looking for co-IP of Saw1. This certainly explains why Mrc1-GFP is not present in the IPs performed at t0 or t1. However, what the authors forgot to mention is how come there is Mrc1-GFP in the Input material at t0 and t1? Is the input not what was subjected to the subsequent IP? If it is, then Mrc1-GFP should be present in the IP at all times. If not, the input used is not appropriate.

Dear Editor,

I am very thankful to the reviewers and their helpful comments. We again revised the manuscript according to the reviewers' comments and performed several additional experiments in response to Reviewers' suggestions. We are confident that we addressed all of the reviewers' comments fully and the manuscript is now greatly improved and highly suitable for publication in Nature Communications.

Reviewer #1

The revised version of the manuscript is significantly improved over the original submission. The manuscript has been completely worked over. It is now clearly articulated and the two main take-home points - the existence of an NER-independent ICL repair pathway in yeast and the isolation of separation of function mutations that selectively abolish the ICL repair and ssDNA annealing functions of Rad1-Rad10 without affecting the repair of UV lesions by NER - are clearly described. The authors also included additional experiments to solidify the claim that the isolated mutant alleles of Rad1 have a defect in the interaction with RPA and SLX4 and address various additional concerns raised in the review.

1. The biochemical data in the manuscript (Fig 4E, S8, S9) is still of insufficient quality for publication. The figure added from another manuscript cited as NAR, under revision (in response to point 12, reviewer 1) does little to address this concern. In looking at the gel provided, I see no additional product formation with increased RPA concentration. The claim that RPA stimulates incision of WT, but not E349K and E706K mutants is simply not convincingly supported by the experiments.

p.14. line 2ff and Fig S8A. I strongly disagree that the "binding of E349K and E706K binding to Y DNA was indistinguishable from WT protein: The binding of the three protein shows very distinct patterns of bands, indicating a different binding mode and/or geometry. Furthermore, I disagree with the assessment that endonuclease efficiency is the same. The values appear different at least for the flap substrate and the values in x-axis of the quantification for S8B and S8C do not represent the values shown in the gel. Fig S8 is simply not publication quality and does not support the claims made by the authors.

p.15, line 2: It is not permissible to cite a paper "in revision" - Reference 72 and the claim associated with it needs to be deleted or the data shown in the supplementary material. See my separate comment on the figure in question above.

We agree with the reviewer that the description of the biochemical data in this section was overly facile. We were most concerned with whether there was a loss in activity, which there does not appear to be. We should, however, have been more nuanced in our description of the mutant data in both the body of the manuscript and in the supplementary materials.

First, we have re-vamped the figures showing the DNA-binding and cleavage experiments. We have shown representative gels and have analyzed the data by scatter plots, which are shown. On the plots, we have also included non-linear regression curves (based on Michaelis-Menten equations) to illustrate the similarities and differences between the catalytic activities. The gels and the plots more closely align, although there are values shown on the plots that do not

appear on the gels shown. We performed these analyses a number of different times, using a wide range of protein concentrations, but they were not necessarily all on the same gels. We have also included an additional representative gel here, to provide further support for our conclusions.

Second, we have provided a more detailed analysis of the *in vitro* results in the main body of the manuscript, on pages 14-15. We discuss the fact that the rad1E349K-Rad10 complex appears to have an altered pattern of DNA binding, although this does not appear to impact its *in vitro* cleavage activity. However, this altered DNA-binding might have an impact on *in vivo* binding, noting that this mutant complex is not efficiently recruited to the recombination intermediate following DSB induction in the cell. We note the rad1E706K-Rad10 appears to have binding pattern similar to Rad1-Rad10, but has enhanced cleavage activity.

Third, we are providing, below, the *correct* version of the figure showing enhanced cleavage of Rad1-Rad10 in the presence of RPA. This is the revised figure for our NAR submission and we regret that the incorrect version was submitted with our previous revision. In addition, we are including a different representative gel that shows increasing cleavage with increasing RPA concentrations, which we have included as part of the supplemental materials for this manuscript.

Figure 11.

Figure 11. rad1R218A disrupts interaction with RPA. **A.** Co-immunoprecipitation experiments from cells. Rad1-HA or rad1R218A-HA was immunoprecipitated with α -HA antibody and associated proteins were probed by Western blot, using α -Rfa1 antibody. **B.** A Representative gel of the endonuclease activity of purified His-Rad1-Rad10 (200 nM) on 3' flap substrates in the absence or presence of increasing

concentrations of RPA. **C.** Representative gel of cleavage activity of His-rad1R218A-Rad10 in the absence or presence of RPA. **D.** Quantification of multiple RPA titration experiments, using multiple different preparations of His-Rad1-Rad10 and RPA. **E.** Quantification of stimulation of cleavage activity by RPA. Note that while the absolute activity of Rad1-Rad10 and rad1R218A-Rad10 in the absence and presence of RPA varies, the relative levels remain consistent. For both panels B and C, the reactions shown were all run on the same gel; some unrelated lanes were cropped out of the images.

(from Eichmiller et al, revision submitted to *NAR*)

2. p5. Lines 7ff. While it possible that ERCC1-XPF has multiple roles in ICL repair, only one of them has been specifically defined and this should be clarified here. Refs 69 Klein -Duwel et al and 71, Abdullah et al, clearly describe a role of ERCC1-XPF in the incision point in ICL repair, and these papers need to be discussed here in the introduction, not just the discussion.

We discuss the role of ERCC1-XPF in the incision of ICL for ICL repair in the introduction and cited both papers in response to the reviewer's comments.

Reviewer #2.

1. Points 1 and 2 - In my opinion, the author's explanation for the viability differences between treatment of G1 or S/G2 cells with ICL-inducing drugs is insufficient. Given that a key conclusion about the relevance of cell cycle stage for ICLR pathway usage is drawn from these genetic experiments, it is important to understand exactly what these experiments reveal. Given that the WT strain is significantly more sensitive to ICL-inducing drugs in G1 than in S/G2, making comparisons about the requirement of certain pathways for repair becomes very complex. At minimum, the authors should choose a range of concentrations of HN2 that causes comparable levels of cell death in G1 and S/G2, in wild-type cells. Otherwise the plots can be rather misleading.

We indeed used three different HN2 doses to interrogate the survival and genetic requirement of cells at different cell cycle stages. AS shown in Fig 1A and S1, the percent survival after 10 μ M HN₂ treatment is not grossly different between G1 (BY4741, 59.7%) and S/G2 (56.2%) cells. At this dose, deletion of MUS81 reduces survival to 36.7% in G1 cells, (1.6-fold as compared to BY4741) or 18.7% in S/G2 cells, (3-fold). Deletion of SAW1 and MUS81 reduces the survival to 12.5% in G1 cells, (4.7-fold) or to 2.0% in G2 cells, (28.1-fold). Therefore, Saw1 and Mus81 appear more important for ICL repair at S/G2. The modest reduction in survival in G1 arrested Mus81 Saw1 cells could be attributed to those that repair ICL lesions at S/G2 after 2 h of G1 arrest and post HN₂ treatment.

Point 3- The staging of cells in the images shown in Figure 2 is, in my opinion, highly subjective. The top DIC image (HN2) in panel B shows a cell that can easily be in mitosis. The lower DIC panel (MMS) shows what could be two independent cells. As suggested before, simple DNA staining with DAPI would have helped solving these staging issues.

In response to the reviewer's suggestions, we now included DAPI staining images in Fig. 2B and Fig. S3.

Point 4- In the main text, line 20, the authors write the following: "Saw1 is present in chromatin fractions pulled down by Mrc1-GFP after HN2 treatment...". In my view, this suggests that what the authors did was to IP Mrc1-GFP in conditions that preserve binding to chromatin, and detect Saw1 in the IPed material. Unfortunately, this is all the information the reader gets to figure out how the experiment was been done. There is also no information in the methods section, except citation of a previous paper, which also cites a previous paper).

In the response to my previous request for clarification of this experiment, in particular to the absence of Mrc1-GFP signal in some of the IPs, the authors write that "the results illustrate the levels of Mrc1 pulled down by chromatin immunoprecipitation. Since most Mrc1 proteins do not associate with chromatin without exogenous DNA damage...the level of Mrc1 in immunoprecipitates is almost undetectable...".

If I understand the authors correctly, what they did was to prepare a chromatin fraction, from which they IPed Mrc1-GFP, looking for co-IP of Saw1. This certainly explains why Mrc1-GFP is not present in the IPs performed at t0 or t1. However, what the authors forgot to mention is how come there is Mrc1-GFP in the Input material at t0 and t1? Is the input not what was subjected to the subsequent IP? If it is, then Mrc1-GFP should be present in the IP at all times. If not, the input used is not appropriate.

The results shown in Fig. 3D are the representative image of chromatin immunoprecipitation of Mrc1-GFP using anti-GFP antibody, followed by immunoblot assay for Saw1 from immunoprecipitates using Saw1 antibody. The goal of this experiment is to test if Saw1 associates with chromatin where replication fork is stalled. Saw1 and Mrc1 do not physically interact with each other. As the control, we included the immunoblot assay for Saw1 and immunoprecipitation with anti-GFP and followed by immunoblot assay with the same antibody for Mrc1-GFP from cell lysates. The control was to make sure that the levels of Mrc1-GFP or Saw1 proteins in the cell lysates were similar between samples and to normalize the amount of Mrc1 proteins bound to chromatin and those of Saw1 among samples. Since Mrc1-GFP does not associate with chromatin without damage or at G1, we did not detect Mrc1-GFP nor Saw1 from chromatin immunoprecipitates (t=0, IP; lane 7 and 10). The levels of Saw1 or Mrc1-GFP in cell lysates were similar (see input, lane 4-6). GFP antibody detects specifically Mrc1-GFP because without Mrc1-GFP expression, we did not detect any signals in the lysates nor ChIP samples (lane 1-3 and 7-9). The levels of Saw1 is not affected by excess expression of Mrc1 in the lysates (compare Saw1 signals in lane 1-3 and 4-6).

REVIEWERS' COMMENTS:

Reviewer #2 (Remarks to the Author):

In my opinion the authors address satisfactorily the key issues raised.
However, I still think the authors could improve a few things without much work:

- 1) improve/include an appropriate explanation of what is done in Figure 3D (both in the main text and in the methods).
- 2) The new DAPI images in Figure 2B are great, but unfortunately show a multinucleated cell for the MMS-treatment condition. It is unclear if this is representative or not.

Response to reviewer's comment.

Reviewer #2 (Remarks to the Author):

In my opinion the authors address satisfactorily the key issues raised. However, I still think the authors could improve a few things without much work:

1) improve/include an appropriate explanation of what is done in Figure 3D (both in the main text and in the methods).

To address the reviewer's comments, we included a sentence in the revision that describes experimental details of Fig. 3D in the Methods.

2) The new DAPI images in Figure 2B are great, but unfortunately show a multinucleated cell for the MMS-treatment condition. It is unclear if this is representative or not.

In response to the reviewer's comments, we now included another DAPI staining image in Fig. 2B.